# OmiAD: One-Step Adaptive Masked Diffusion Model for Multi-class Anomaly Detection via Adversarial Distillation

**Yaoxuan Feng** [1]  **Wenchao Chen** [1]  **Yuxin Li** [1]  **Bo Chen** [1]  **Yubiao Wang** [1]  **Zixuan Zhao** [1]  **Hongwei Liu** [1]
**Mingyuan Zhou** [2]

## Abstract

Diffusion models have demonstrated outstanding performance in industrial anomaly detection. However, their iterative denoising nature results in slow inference speed, limiting their practicality for real-time industrial deployment. To address this challenge, we propose OmiAD, a **O**ne-step **m**asked d**i**ffusion model for multi-class **A**nomaly **D**etection, derived from a well-designed multi-step adaptive masked diffusion model (AMDM) and compressed using adversarial score distillation (ASD). OmiAD first introduces AMDM, equipped with an adaptive masking strategy that dynamically adjusts masking patterns based on noise levels and encourages the model to reconstruct anomalies as normal counterparts by leveraging broader context, to reduce the pixel-level shortcut reliance. Then, ASD is developed to compress the multi-step diffusion process into a single-step generator by score distillation and incorporating a shared-weight discriminator effectively reusing parameters while significantly improving both inference efficiency and detection performance. The effectiveness of OmiAD is validated on four diverse datasets, achieving state-of-the-art performance across seven metrics while delivering a remarkable inference speedup. Code is available at https://github.com/luolundashu/OmiAD

## 1. Introduction

Anomaly detection plays a pivotal role in manufacturing defect detection (Bergmann et al., 2019), medical image analysis (Fernando et al., 2021), and video surveillance (Ramachandra et al., 2020). In industrial environments, due to the scarcity of anomalous samples, unsupervised anomaly detection has gained significant attention for its capability to learn the normal data distribution and detect deviations indicative of anomalies (You et al., 2022; Lu et al., 2023; He et al., 2024b). Furthermore, industrial anomaly detection often involves objects spanning diverse categories with distinct normal patterns. Such scenarios require models capable of generalizing across different classes without necessitating extensive fine-tuning. As such, developing a robust unsupervised multi-class anomaly detection framework is crucial to achieving consistent and efficient production workflows in real-world industrial scenarios.

Under the paradigm of unsupervised multi-class anomaly detection, diffusion-based generative models (Sohl-Dickstein et al., 2015; Song et al., 2020; Ho et al., 2020) have garnered substantial interest for their ability to handle high-dimensional data and model complex distributions. These models operate by progressively denoising a sample from random noise, enabling them to reconstruct intricate data patterns that traditional methods often struggle to capture. This generative process makes diffusion models are particularly effective at learning the distribution of normal data and identifying deviations that signal anomalies, making them a promising approach for unsupervised multi-class anomaly detection tasks (Yin et al., 2023; He et al., 2024b).

Several diffusion-based approaches have been proposed for anomaly detection, with a primary focus on generating impactful conditional embeddings derived from abnormal inputs (Mousakhan et al., 2023; Yin et al., 2023; He et al., 2024b; Fučka et al., 2025; Yao et al., 2025). These embeddings are then fed into the denoising network to guide the reverse process within the diffusion model. For instance, DADD (Mousakhan et al., 2023) conditions the denoising process on the original input, enabling the model to progressively remove noise and accurately reconstruct the normal distribution. Similarly, DiAD (He et al., 2024b) leverages pixel-level semantic information as conditional inputs to guide the denoising process. While these methods enhance reconstruction quality, they often overlook a critical limitation in reconstruction-based anomaly detection, known as

---

[1]National Key Laboratory of Radar Signal Processing, Xidian University, Xi'an, 710071, China [2]McCombs School of Business, The University of Texas at Austin, Austin, TX 78712, USA. Correspondence to: Wenchao Chen <chenwenchao@xidian.edu.cn>.

*Proceedings of the 42nd International Conference on Machine Learning*, Vancouver, Canada. PMLR 267, 2025. Copyright 2025 by the author(s).

"identical shortcut" (Gong et al., 2019; You et al., 2022), which arises when models over-rely on local, pixel-level features, unintentionally preserving abnormal features during reconstruction. As a result, these models prioritize local patterns to minimize reconstruction errors, unintentionally reproducing anomalies in the output.

Moreover, the iterative nature of diffusion models poses a significant challenge for real-time deployment due to their reliance on a stepwise denoising process, where each iteration gradually removes noise to reconstruct the original data. This process requires multiple forward passes through the network, leading to high computational overhead and slow inference speeds. While methods based on faster solvers for stochastic differential equations (SDE) or ordinary differential equations (ODE) have successfully reduced the number of inference steps to around a dozen (Zhang & Chen, 2022; Karras et al., 2022; Lu et al., 2022; Song & Dhariwal, 2023), it still falls short of meeting the stringent real-time requirements of industrial applications.

These challenges highlight the need for a new approach that can reduce "identical shortcut" and improve inference efficiency in diffusion-based anomaly detection. To this end, we propose a **O**ne-step adaptive **m**asked d**i**ffusion model for **A**nomaly **D**etection, named OmiAD, which is a novel unsupervised framework that addresses both issues through two key innovations: an **A**daptive **M**asking strategy (AM), which mitigates shortcut reliance by dynamically encouraging broader contextual understanding based on noise levels, and **A**dversarial **S**core **D**istillation (ASD), which compresses the multi-step diffusion process into a single inference step by distilling the knowledge of training data encapsulated within the score-estimation network of a pretrained diffusion model. During this distillation process, OmiAD synchronously employs the encoder module of the student score network as a shared-weight discriminator to perform adversarial distillation, eliminating the need for additional parameters (Zhou et al., 2024a). This Diffusion GAN-based adversarial loss enables the model to distinguish between real and generated samples by aligning the noisy distributions at any timestep (Wang et al., 2022), ensuring that the generative distribution closely matches the clean data distribution. By integrating these components, OmiAD achieves a more efficient and streamlined training framework, improving anomaly detection accuracy while maintaining computational efficiency and simplicity, making it highly applicable to real-world scenarios.

The main contributions of our work are summarized as follows:

- We propose a novel Adaptive Masked Diffusion Model (AMDM) that utilizes an AM strategy to guide the model in reconstructing anomalies into their normal counterparts, effectively mitigating the "identical short-

cut" problem and enhancing the model's ability to capture global context.

- We adapt adversarial score distillation for anomaly detection and develop OmiAD, which compresses the multi-step diffusion process into a single inference step, significantly improving efficiency. To further reduce redundancy, OmiAD incorporates a shared-weight discriminator that reuses parameters, enabling a more compact and efficient training process.

- OmiAD is validated on four datasets, achieving state-of-the-art performance across seven metrics while delivering remarkable speed-ups.

## 2. Background

### 2.1. Multi-class Unsupervised Anomaly Detection

Multi-class unsupervised anomaly detection has garnered increasing attention for its ability to create a unified model that detects anomalies across multiple categories when only normal data is available (Li et al., 2021; Pirnay & Chai, 2022; Chen et al., 2022; You et al., 2022). It is based on the hypothesis that reconstruction models trained on normal samples excel in normal regions but struggle with anomalous ones (Deng & Li, 2022; Liu et al., 2023; Mousakhan et al., 2023; Zhang et al., 2023b; He et al., 2024a). Building upon this hypothesis, recent works have introduced innovative methods to enhance anomaly detection performance. UniAD (You et al., 2022) integrates Neighbor Masked Attention and Layer-wise Query Decoder mechanisms, effectively mitigating the "identical shortcut" problem and improving anomaly detection performance. HVQ-Trans (Lu et al., 2023) utilizes vector quantization to prevent the model from learning "identical shortcut," thereby enhancing the robustness of anomaly detection. DiAD (He et al., 2024b) explores an anomaly detection framework based on diffusion models, introducing a semantic-guided network to ensure the consistency of reconstructed image semantics.

### 2.2. Diffusion Distillation

Diffusion models are celebrated for their ability to generate high-fidelity and diverse data, but they are often constrained by the large number of sampling steps required for refinement. To address this limitation, diffusion distillation techniques have emerged as an effective solution, compressing multi-step diffusion processes into single-step or few-step generators (Salimans & Ho, 2022; Song et al., 2023).

Recent advances in diffusion distillation are primarily driven by distribution matching under noise corruption. These methods aim to align the distribution of generated data—after adding noise at various levels—with that of real data. An early attempt at this idea was Diffusion GAN (Wang

et al., 2022), which matched noisy data distributions directly but required access to real data samples. In contrast, diffusion distillation can eliminate this requirement by representing the noisy data distribution using the estimated score function from a pretrained diffusion model.

To measure the discrepancy between the noisy data and generated distributions in score space, these methods typically use either the KL divergence—as seen in Score Distillation Sampling (SDS) (Poole et al., 2022), Variational Score Distillation (Wang et al., 2023), Diff-Instruct (Luo et al., 2024a), and Distribution Matching Distillation (DMD) (Yin et al., 2024b)—or the Fisher divergence, employed in Score identity Distillation (SiD) (Zhou et al., 2024b) and its variants (Zhou et al., 2024a; 2025; Luo et al., 2024b; Huang et al., 2024). SiD leverages three score-related identities to construct a novel loss function without relying on real data, enabling state-of-the-art performance in single-step generation tasks.

To further improve both generation efficiency and quality—particularly by correcting inaccuracies in the teacher diffusion model—adversarial diffusion distillation methods (Sauer et al., 2025; Yin et al., 2024a; Zhou et al., 2024a) have shown great promise by integrating diffusion models with GAN-style training (Goodfellow et al., 2014) or Diffusion GAN frameworks (Wang et al., 2022). For example, ADD (Sauer et al., 2025) introduces an adversarial loss into the distillation process, significantly boosting both sample fidelity and inference speed. SiDA (Zhou et al., 2024a) further extends SiD by incorporating adversarial loss across varying noise levels, enabling finer discrimination between real and generated samples during distillation.

## 3. Methodology

To harness the generative capabilities of diffusion models for anomaly detection while mitigating the inefficiencies of their multi-step generation process, we first design the AMDM, which incorporates AM to enhance robustness and efficiency across diverse scenarios. Then, we introduce ASD to compress AMDM into a single-step model, significantly improving inference speed and detection performance. OmiAD seamlessly integrates AMDM with ASD, achieving efficient and high-performing multi-class anomaly detection.

### 3.1. Preliminaries

Diffusion models, particularly the widely adopted Denoising Diffusion Probabilistic Model (DDPM) (Ho et al., 2020), consist of two key processes: a forward diffusion process and a reverse diffusion process. The forward diffusion process incrementally corrupts the input vector $\boldsymbol{x}_0$ by adding noise over $T$ steps, progressively transforming it into a

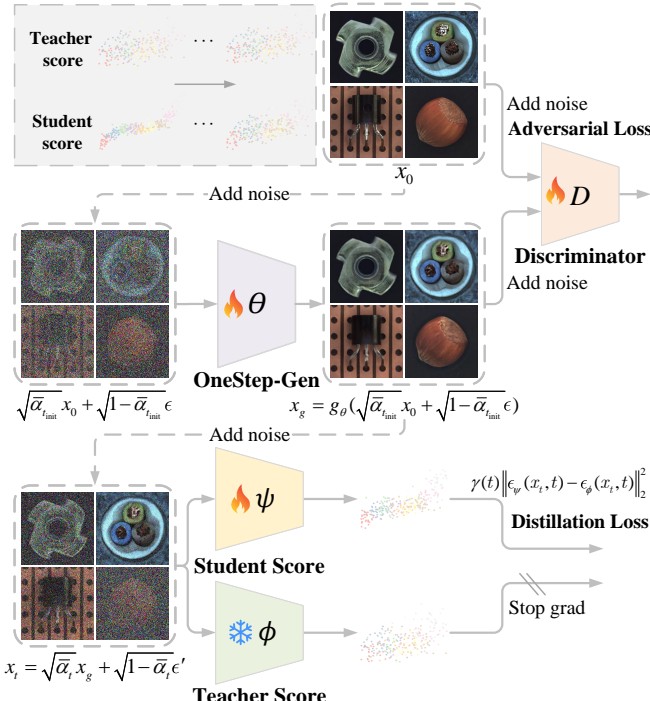

Figure 1. Overview of the ASD framework. ASD comprises four main components: One Step Generator (OneStep-Gen), Teacher Score, Student Score, and Discriminator, which collaboratively enable accurate single-step generation and robust anomaly detection. The upper-left section illustrates the alignment between the Teacher and Student Scores, where the Student Score progressively adapts to the Teacher Score, thereby optimizing the One Step Generator. For a detailed explanation, please refer to Section 3.3.

Gaussian noise vector $\boldsymbol{x}_T$:

$$q(\boldsymbol{x}_{1:T} \mid \boldsymbol{x}_0) := \prod_{t=1}^{T} q(\boldsymbol{x}_t \mid \boldsymbol{x}_{t-1}),$$
$$q(\boldsymbol{x}_t \mid \boldsymbol{x}_{t-1}) := \mathcal{N}(\sqrt{1-\beta_t}\boldsymbol{x}_{t-1}, \beta_t \boldsymbol{I}) \tag{1}$$

where $\beta_t$ represents a positive constant denoting the noise level. In practical applications, we directly sample $\boldsymbol{x}_t$ given $\boldsymbol{x}_0$ as the following:

$$q(\boldsymbol{x}_t \mid \boldsymbol{x}_0) = \mathcal{N}(\sqrt{\bar{\alpha}_t}\boldsymbol{x}_0, (1-\bar{\alpha}_t)\boldsymbol{I}) \tag{2}$$

where $\alpha_t := 1-\beta_t$ and $\bar{\alpha}_t := \prod_{t=1}^{T} \alpha_t$. The reverse process involves denoising $\boldsymbol{x}_t$ back to $\boldsymbol{x}_0$:

$$p_\theta(\boldsymbol{x}_{0:T}) := p(\boldsymbol{x}_T) \prod_{t=1}^{T} p_\theta(\boldsymbol{x}_{t-1} \mid \boldsymbol{x}_t),$$
$$p_\theta(\boldsymbol{x}_{t-1} \mid \boldsymbol{x}_t) := \mathcal{N}(\boldsymbol{\mu}_\theta(\boldsymbol{x}_t, t), \boldsymbol{\sigma}_\theta(\boldsymbol{x}_t, t)) \tag{3}$$

In DDPM, $p_\theta(\boldsymbol{x}_{t-1} \mid \boldsymbol{x}_t)$ is defined as:

$$\boldsymbol{\mu}_\theta(\boldsymbol{x}_t, t) = \frac{1}{\sqrt{\alpha_t}}(\boldsymbol{x}_t - \frac{\beta_t}{\sqrt{1-\bar{\alpha}_t}}\boldsymbol{\epsilon}_\theta(\boldsymbol{x}_t, t)),$$
$$\boldsymbol{\sigma}_\theta(\boldsymbol{x}_t, t) = (\bar{\beta}_t)^{1/2}, \tag{4}$$
$$\text{if } t=1: \bar{\beta}_t = \beta_1, \quad \text{else}: \bar{\beta}_t = \frac{1-\bar{\alpha}_{t-1}}{1-\bar{\alpha}_t}\beta_t$$

where the $\epsilon_\theta$ is denoising function and which can be trained by solving the following optimization problem:

$$\min_\theta \mathcal{L}(\theta) := \mathbb{E}_{q(x_t|x_0,t)}[\gamma(t) \, \|\epsilon - \epsilon_\theta(\boldsymbol{x}_t, t)\|_2^2] \quad (5)$$

where $\gamma(t)$ is a time-dependent weight that adjusts the contribution of different timesteps during training, ensuring an appropriate balance across noise levels.

### 3.2. Adaptive Masked Diffusion Model

Existing diffusion-based anomaly detection models often suffer from the "identical shortcut" problem. To address this issue, we propose the AMDM, which integrates AM to dynamically adjust the masking probability $p(t)$ based on the noise level during the diffusion process. This mechanism adaptively balances preserving local details at low noise levels and modeling broader context at high noise levels. By progressively increasing $p(t)$ with noise, AM enhances global context modeling while retaining critical features, thereby improving robustness and efficiency in anomaly detection across diverse scenarios.

The original image is first processed by EfficientNet to extract its feature representation, denoted as $\boldsymbol{x_0}$, which is situated in the input feature space. Subsequently, a binary mask $\mathbf{M}(t)$ is applied to $\boldsymbol{x_0}$, producing the masked feature representation $\boldsymbol{x_m}$:

$$\boldsymbol{x_m} = \boldsymbol{x_0} \odot \mathbf{M}(t), \quad (6)$$

where $\odot$ denotes element-wise multiplication. The masking probability $p(t)$ is dynamically adjusted based on the forward diffusion timestep $t$, ensuring that the masking process is synchronized with the diffusion process. Specifically, $p(t)$ is defined as:

$$p(t) = p_{\min} + (p_{\max} - p_{\min}) \cdot \left(\frac{t}{T}\right)^k \quad (7)$$

where $p_{\min}$ and $p_{\max}$ represent the minimum and maximum masking probabilities, $T$ is the total number of diffusion timesteps, and $k$ controls the masking growth rate. The binary mask $\mathbf{M}(t)$ is sampled as:

$$\mathbf{M}(t) \sim \text{Bernoulli}(p(t)) \quad (8)$$

The forward diffusion timestep $t$ is consistent with the masking process, and Gaussian noise is added to the masked feature $\boldsymbol{x_m}$ at the same timestep $t$. At each timestep $t$, the noisy feature $\boldsymbol{x_m}^t$ is computed as:

$$\boldsymbol{x_m}^t = \sqrt{\bar{\alpha}_t} \cdot \boldsymbol{x_m} + \sqrt{1 - \bar{\alpha}_t} \cdot \boldsymbol{\epsilon}, \quad \boldsymbol{\epsilon} \sim \mathcal{N}(0, \mathbf{I}) \quad (9)$$

where $\bar{\alpha}_t$ represents the variance schedule. The noisy feature $\boldsymbol{x_m}^t$, along with the timestep $t$, is passed through the model $\theta$ to predict the denoised feature $\hat{\boldsymbol{x_0}}$:

$$\hat{\boldsymbol{x_0}} = \frac{1}{\sqrt{\bar{\alpha}_t}}\boldsymbol{x_m}^t - \frac{\sqrt{1 - \bar{\alpha}_t}}{\sqrt{\bar{\alpha}_t}}\boldsymbol{\epsilon}_\theta(\mathbf{x_m}^t, t) \quad (10)$$

where $\epsilon_\theta(\boldsymbol{x_m}^t, t)$ is the noise predicted by the model $\theta$. By using the timestep $t$ for both masking adjustment and noise addition, the model ensures consistency throughout the forward diffusion process, enabling accurate reconstruction of the original feature $\boldsymbol{x_0}$ by estimating and removing the noise component.

The reconstructed feature $\hat{\boldsymbol{x_0}}$ is compared with the ground truth $\boldsymbol{x_0}$.

$$\mathcal{L} = \gamma(t)\|\hat{\boldsymbol{x_0}} - \boldsymbol{x_0}\|_2^2 \quad (11)$$

By incorporating AM into the diffusion process, the proposed AMDM significantly strengthens global context modeling, effectively mitigating reliance on shortcut features. This enhancement improves the model's accuracy in anomaly detection across diverse scenarios while ensuring robustness and efficiency.

### 3.3. One-step Adaptive Masked Diffusion Model

As a diffusion-based method, AMDM suffers from slow inference due to its iterative denoising process. To address this limitation, we adapt the SiD method proposed in Zhou et al. (2024b) and SiDA method proposed in Zhou et al. (2024a) to develop OmiAD, a novel one-step adaptive masked diffusion model. The ASD framework used by OmiAD is illustrated in Fig. 1. ASD comprises four components: One Step Generator (OneStep-Gen), Teacher Score, Student Score, and Discriminator. The Teacher Score $\epsilon_\phi$, derived from AMDM, leverages its global contextual modeling for anomaly detection. The One Step Generator $g_\theta$ performs single-step generation to reconstruct $\boldsymbol{x_g}$, which, along with the real sample $\boldsymbol{x_0}$, is perturbed with the same noise and fed into the Discriminator $D$ to align noisy distributions. Additionally, $\boldsymbol{x_g}$ is further perturbed with noise and processed through the Teacher Score $\epsilon_\phi$ and Student Score $\epsilon_\psi$. The loss is computed by aligning the Student Score with the Teacher Score, thereby facilitating the optimization of the One Step Generator. These four components within ASD are integrated into the score distillation and adversarial optimization processes, enabling OmiAD to achieve not only efficient single-step generation but also enhanced anomaly detection capabilities.

#### 3.3.1. SCORE DISTILLATION

Our distillation process leverages the One Step generator $g_\theta$ to produce data that closely matches the real data distribution. By training the student score $\epsilon_\psi$ on this data, it learns to replicate the behavior of the teacher score $\epsilon_\phi$. Once the distillation is complete, $g_\theta(\boldsymbol{x_0})$ can perform anomaly detection in a single step, by passing the need for multi-step inference while retaining the teacher model's performance.

The student score $\epsilon_\psi$ is trained using data generated by the

single-step generator $g_\theta$, formulated as:

$$\boldsymbol{x_g} = g_\theta(\sqrt{\bar{\alpha}_{t_{\text{init}}}}\boldsymbol{x_0} + \sqrt{1 - \bar{\alpha}_{t_{\text{init}}}}\boldsymbol{\varepsilon}) \tag{12}$$

where $\boldsymbol{x_0}$ represents a normal training sample, and $\boldsymbol{x_g}$ is the output reconstructed by the generator. We denote the distribution of $\boldsymbol{x_g}$ as $p_\theta(\boldsymbol{x_g})$. The timestep $t_{\text{init}}$ represents a fixed noise level used during the generation of $\boldsymbol{x_g}$. The reconstructed samples $\boldsymbol{x_g}$ are then used to train the student score $\epsilon_\psi$, defined as:

$$\mathbb{E}_{q(x_t|x_g,t)p_\theta(x_g)}\left[\left\|\boldsymbol{\epsilon}_\psi(\sqrt{\bar{\alpha}_t}\boldsymbol{x_g} + \sqrt{1 - \bar{\alpha}_t}\boldsymbol{\epsilon}, t) - \boldsymbol{\epsilon}\right\|_2^2\right] \tag{13}$$

Assuming we have access to $\epsilon_{\psi^*(\theta)}$, the optimal student score that minimizes Equation 13, we follow SiD to define a Fisher divergence-based distillation loss to optimize $\boldsymbol{g_\theta}$, expressed as

$$\mathcal{L}_\theta = \mathbb{E}_{q(x_t|x_g,t)p_\theta(x_g)}\left[\gamma(t)\left\|\boldsymbol{\epsilon}_{\psi^*(\theta)}(\boldsymbol{x_t}, \boldsymbol{t}) - \boldsymbol{\epsilon}_\phi(\boldsymbol{x_t}, \boldsymbol{t})\right\|_2^2\right] \tag{14}$$

While Equation 14 is intractable to solve due to the unknown $\psi^*(\theta)$ and its gradient, we follow SiD and adopt an alternating optimization strategy. This approach alternates between optimizing $\psi$ using SGD with Equation 13 and optimizing $\theta$ using SGD with the following loss:

$$\mathcal{L}_\theta = \mathbb{E}_{q(x_t|x_g,t)p_\theta(x_g)}\Big[\gamma(t)\Big(\Delta_{\psi,\phi}(\boldsymbol{x_t})\big(\boldsymbol{\epsilon} - \boldsymbol{\epsilon}_\phi(\boldsymbol{x_t}, \boldsymbol{t})\big)$$
$$- \alpha\|\Delta_{\psi,\phi}(\boldsymbol{x_t})\|_2^2\Big)\Big] \tag{15}$$

where $\Delta_{\psi,\phi}(\boldsymbol{x_t})$ represents the discrepancy between the teacher and student scores:

$$\Delta_{\psi,\phi}(\boldsymbol{x_t}) = \boldsymbol{\epsilon}_\psi(\boldsymbol{x_t}, \boldsymbol{t}) - \boldsymbol{\epsilon}_\phi(\boldsymbol{x_t}, \boldsymbol{t}) \tag{16}$$

Through this SiD-based score distillation process, the multi-step diffusion mechanism of AMDM is compressed into a single inference step. This is accomplished by extracting and distilling the knowledge from the training data encapsulated within the score-estimation network of a pretrained diffusion model. The derivation of this formulation, which follows the original approach in SiD (Zhou et al., 2024b; 2025) but is adapted to the DDPM-style teacher model used in this paper, is provided in Appendix C.

### 3.3.2. ADVERSARIAL OPTIMIZATION IN DISTILLATION

Through OmiAD, single-step anomaly detection is achieved via distillation. Following SiDA (Zhou et al., 2024a), we enhance performance by integrating a Diffusion GAN (Wang et al., 2022) based adversarial loss. As in SiDA, OmiAD repurposes the encoder module of the student score network as a shared-weight discriminator, eliminating the need for additional parameters. This adversarial loss enables the model to distinguish between real and generated samples by aligning the noisy distributions at any timestep $t$, ensuring that

the generative distribution closely matches the clean data distribution, thereby improving anomaly detection accuracy.

To train the encoder module of $\epsilon_\psi$, which also functions as the discriminator, we define its role in detail. Specifically, the discriminator corresponds to the encoder component of the network. For an input $\boldsymbol{x_t}$, which is a noisy sample, the encoder module processes the input and generates the final layer output of the discriminator with dimensions $(H, W, C)$, where $H$ and $W$ represent the spatial dimensions, and $C$ denotes the number of channels, we compute the mean along the channel dimension, resulting in a feature map of size $(H, W)$.

This mechanism enables the discriminator to capture subtle differences between real and generated (reconstructed) samples. Particularly in noisy scenarios, the discriminator's feature map highlights local responses that amplify the anomaly signals between the generated and real samples.

To this end, we define the discriminator loss as:

$$L_\psi^{\text{adv}} = \frac{1}{W'H'} \sum_{i'=1}^{W'} \sum_{j'=1}^{H'} \Big[\ln D(\boldsymbol{y_t})[i', j'] + \ln\big(1 - D(\boldsymbol{x_t})[i', j']\big)\Big] \tag{17}$$

where $\boldsymbol{x_t}$ represents the generated (reconstructed) sample, and $\boldsymbol{y_t}$ denotes the real sample. $D$ is discriminator. $D(\boldsymbol{y_t})[i', j']$ and $D(\boldsymbol{x_t})[i', j']$ denote the responses at the $(i', j')$-th position in the discriminator maps for real and generated samples, respectively. This formulation ensures the discriminator learns to distinguish between real and fake samples by optimizing its responses across the spatial dimensions $W'$ and $H'$.

Next, we leverage the discriminator to optimize $\theta$, with the adversarial loss defined as:

$$\mathcal{L}_\theta^{\text{adv}} = \frac{1}{W'H'} \sum_{i=1}^{W'} \sum_{j=1}^{H'} D(\boldsymbol{x_t})[i, j] \tag{18}$$

This formulation ensures the discriminator learns to distinguish between real and fake samples by optimizing its responses across the spatial dimensions $W'$ and $H'$.

In summary, the adversarial distillation process is completed with minimal computational overhead due to the reuse of the encoder module as the discriminator. This efficient design not only ensures that distillation incurs almost no additional computational cost but also enhances the model's sensitivity and capability in anomaly detection. The detailed ASD algorithm is provided in Algorithm 1.

*Table 1.* Quantitative Results on different AD datasets for multi-class setting.

| Dateset | Method | Image-level | | | Pixel-level | | | | mAD |
|---------|--------|--------|------|--------|--------|------|--------|--------|------|
| | | AU-ROC | AP | F1_max | AU-ROC | AP | F1_max | AU-PRO | |
| MVTec-AD | RD4AD | 94.6 | 96.5 | 95.2 | 96.1 | 48.6 | 53.8 | 91.1 | 82.3 |
| | UniAD | 96.5 | 98.8 | 96.2 | 96.8 | 43.4 | 49.5 | 90.7 | 81.7 |
| | SimpleNet | 95.3 | 98.4 | 95.8 | 96.9 | 45.9 | 49.7 | 86.5 | 81.2 |
| | DeSTSeg | 89.2 | 95.5 | 91.6 | 93.1 | 54.3 | 50.9 | 64.8 | 77.1 |
| | DiAD | 97.2 | 99.0 | 96.5 | 96.8 | 52.6 | 55.5 | 90.7 | 84.0 |
| | HVQ-Trans | 98.0 | 99.5 | 97.5 | 97.3 | 48.2 | 53.3 | 91.4 | 83.6 |
| | AMDM(Ours) | 98.4 | 99.0 | 97.4 | 97.5 | 51.5 | 56.1 | 92.6 | 84.6 |
| | OmiAD (Ours) | **98.8** | **99.7** | **98.5** | **97.7** | 52.6 | **56.7** | **93.2** | **85.3** |
| VisA | RD4AD | 92.4 | 92.4 | 89.6 | 98.1 | 38.0 | 42.6 | **91.8** | 77.8 |
| | UniAD | 88.8 | 90.8 | 85.8 | 98.3 | 33.7 | 39.0 | 85.5 | 74.6 |
| | SimpleNet | 87.2 | 87.0 | 81.8 | 96.8 | 34.7 | 37.8 | 81.4 | 72.4 |
| | DeSTSeg | 88.9 | 89.0 | 85.2 | 96.1 | 39.6 | 43.4 | 67.4 | 72.8 |
| | DiAD | 86.8 | 88.3 | 85.1 | 96.0 | 26.1 | 33.0 | 75.2 | 70.1 |
| | HVQ-Trans | 93.2 | 92.8 | 87.6 | 98.7 | 35.0 | 39.6 | 86.3 | 76.2 |
| | AMDM(Ours) | 94.8 | 95.6 | 91.1 | 98.8 | 39.8 | 43.5 | 88.4 | 78.9 |
| | OmiAD (Ours) | **95.3** | **96.0** | **91.2** | **98.9** | **40.4** | **44.1** | 89.2 | **79.3** |
| MPDD | RD4AD | 84.1 | 83.2 | 84.1 | 98.1 | 35.2 | 38.7 | 93.4 | 73.8 |
| | UniAD | 82.2 | 87.1 | 85.1 | 95.1 | 18.9 | 25.0 | 81.9 | 67.9 |
| | SimpleNet | 90.6 | 94.1 | 89.7 | 97.1 | 33.6 | 35.7 | 90.0 | 75.8 |
| | DeSTSeg | 93.0 | 95.1 | 90.6 | 94.1 | 33.2 | 37.6 | 59.8 | 71.9 |
| | DiAD | 74.6 | 82.1 | 82.5 | 93.0 | 15.9 | 21.2 | 78.4 | 64.0 |
| | HVQ-Trans | 86.5 | 88.1 | 85.8 | 96.7 | 27.6 | 31.4 | 86.9 | 71.9 |
| | AMDM(Ours) | 93.3 | 94.9 | 90.7 | 98.4 | 36.8 | 41.6 | 93.5 | 78.4 |
| | OmiAD (Ours) | **93.7** | **95.5** | **90.9** | **98.6** | **37.6** | **42.3** | **94.0** | **78.9** |
| Real-IAD | RD4AD | 82.4 | 79.0 | 73.9 | 97.3 | 25.0 | 32.7 | 89.6 | 68.6 |
| | UniAD | 83.0 | 80.9 | 74.3 | 97.3 | 21.1 | 29.2 | 86.7 | 67.5 |
| | SimpleNet | 57.2 | 53.4 | 61.5 | 75.7 | 2.8 | 6.5 | 39.0 | 42.3 |
| | DeSTSeg | 82.3 | 79.2 | 73.2 | 94.6 | **37.9** | 41.7 | 40.6 | 64.2 |
| | DiAD | 75.6 | 66.4 | 69.9 | 88.0 | 2.9 | 7.1 | 58.1 | 52.6 |
| | HVQ-Trans | 86.6 | 84.9 | 79.4 | 98.0 | 27.6 | 34.4 | 88.7 | 71.4 |
| | AMDM(Ours) | 89.8 | 87.7 | 81.9 | 98.6 | 36.5 | 41.4 | 92.6 | 75.5 |
| | OmiAD (Ours) | **90.1** | **88.6** | **82.8** | **98.9** | 37.7 | **42.6** | **93.1** | **76.3** |

## 4. Experiment

### 4.1. Experiments Setup

**Dataset:** Four datasets are utilized in our paper: (1) **MVTec-AD** (Bergmann et al., 2019) simulates real-world industrial production scenarios with high-resolution images specifically designed for unsupervised anomaly detection. (2) **VisA** (Zou et al., 2022), a large dataset with 9,621 normal and 1,200 anomalous images spanning 12 object types and diverse anomalies. (3) **MPDD** (Jezek et al., 2021), focusing on detecting defects in six classes of metal parts during fabrication. (4) **Real-IAD** (Wang et al., 2024) comprises 150,000 images across 30 distinct categories, offering a comprehensive large-scale benchmark for anomaly detection. More details about the datasets can be found in Appendix D.

**Evaluation metrics:** For anomaly detection and segmentation, we report the Area Under the Receiver Operating Characteristic Curve (AU-ROC), Average Precision (AP) (Zavrtanik et al., 2021), and F1-score-max (F1_max) (Zou et al., 2022). Additionally, for anomaly segmentation, we also report the Area Under the Per-Region-Overlap (AU-PRO) (Bergmann et al., 2020). To provide a comprehensive as-

sessment of the model's performance, we further calculate the mean value of the above seven metrics, referred to as mAD (Zhang et al., 2023a).

**Baselines:** We extensively compare our model with 6 baseline methods under unsupervised multi-class settings, including the unified state-of-the-art method HVQ-Trans (Lu et al., 2023) and the diffusion-based approach DiAD (He et al., 2024b). More details about the baseline methods can be found in Appendix E.

**Implementation details:** For AMDM, the number of timesteps is $T = 1000$, with a linear noise schedule ($\beta_1 = 10^{-4}$, $\beta_T = 0.02$) as in Ho et al. (2020). The mask probability is controlled by $p_{\min} = 0.1$, $p_{\max} = 0.4$ and $k = 2$, while the weighting factor $\alpha$ in adversarial distillation is set to 1. More details are in Appendix F.

### 4.2. Comparison with SoTAs on Different AD datasets

As shown in Table 1, OmiAD achieves state-of-the-art performance across four widely used anomaly detection datasets, highlighting its robustness and versatility. Built upon its teacher model, AMDM, which leverages global

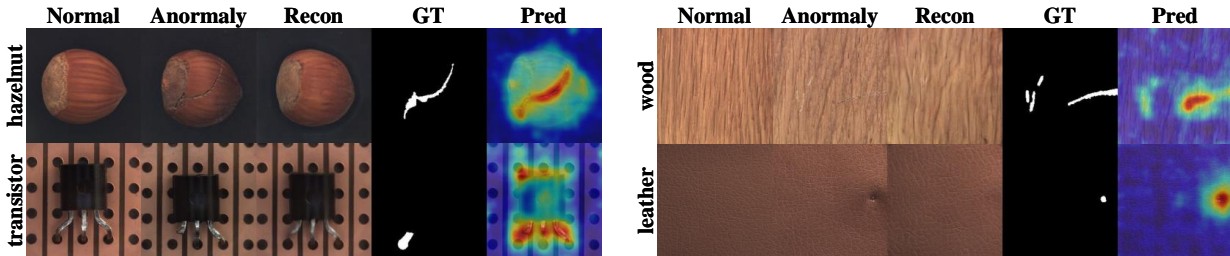

*Figure 2.* Qualitative results for anomaly localization on MVTec-AD. From left to right: normal sample as the reference, anomaly sample, our reconstruction, ground-truth, and our predicted anomaly map.

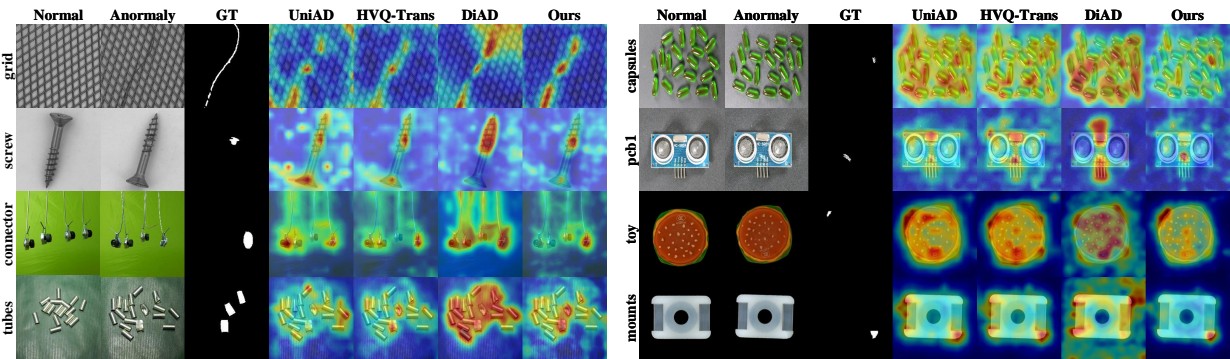

*Figure 3.* Qualitative comparison of pixel-level anomaly segmentation results across four datasets. From left to right: normal sample as the reference, anomaly sample, ground truth (GT), predicted anomaly maps by UniAD, HVQ-Trans, DiAD, and ours.

contextual information to secure the second-highest performance across four datasets, OmiAD demonstrates its capability to excel in both anomaly detection and segmentation tasks.

On the MVTec-AD dataset, OmiAD achieves impressive scores of 98.8/99.7/98.5 for AU-ROC/AP/F1_max at the image level and 97.7/52.6/56.7/93.2 for AU-ROC/AP/F1_max/PRO at the pixel level, surpassing DiAD by 1.5% and HVQ-Trans by 2.0% in the overall mAD metric. On the VisA dataset, OmiAD outperforms DiAD by 13.1% and HVQ-Trans by 4.1% in mAD. For the MPDD dataset, which features industrial metal parts with subtle, localized defects, OmiAD achieves a significant 23.3% improvement over DiAD and 9.7% over HVQ-Trans in mAD. On the Real-IAD dataset, a large-scale benchmark of 150K images across 30 categories, OmiAD demonstrates exceptional performance, surpassing DiAD by 45.1% and HVQ-Trans by 6.9% in mAD, further underscoring its scalability and effectiveness in real-world applications. Comprehensive per-category results for each dataset are provided in Appendix G.

### 4.3. Quantitative Results of OmiAD

To demonstrate the capability of modeling normal distributions, we visualize the generated results. As illustrated in Fig. 2, OmiAD effectively reconstructs anomalies into their corresponding normal samples, accurately localizing anomalous regions through reconstruction differences. Specifically, it handles both object anomalies (Left) and texture damages (Right) with remarkable precision, thereby showcasing its robustness in capturing subtle and diverse defect patterns across various scenarios.

Furthermore, Fig. 3 highlights OmiAD's superior pixel-level anomaly segmentation capabilities. Compared to UniAD, HVQ-Trans, and DiAD, OmiAD achieves significantly more accurate segmentation with minimal bias. This advantage can be attributed to its ability to incorporate global contextual information, which enables it to avoid reliance on localized features. Consequently, OmiAD delivers robust and consistent segmentation performance across a wide range of anomaly types. For additional qualitative results and analysis, please refer to Appendix J.

### 4.4. Inference Speed Comparison

We evaluate the inference speed of OmiAD against Transformer-based, Reconstruction-based, and Diffusion-based methods on four anomaly detection datasets, as shown in Table 2. All evaluations are conducted with a batch size of 64 under identical hardware settings.

OmiAD achieves unparalleled inference efficiency across all

*Table 2.* Average inference time (in seconds) for different models on various datasets with batch size 64.

| Dataset | Transformer-based | | Reconstruction-based | | Diffusion-based | | | Ours |
|---|---|---|---|---|---|---|---|---|
| | HVQ-Trans | UniAD | RD4AD | ReContrast | DDAD | DiAD | TransFusion | |
| MVTec-AD | 0.1795 | 0.1891 | 0.2075 | 0.2383 | 6.0067 | 15.662 | 15.904 | **0.0254** |
| VisA | 0.2018 | 0.2198 | 0.2063 | 0.2342 | 6.0260 | 16.446 | 15.975 | **0.0283** |
| MPDD | 0.1620 | 0.1723 | 0.2106 | 0.2314 | 5.9247 | 16.327 | 16.002 | **0.0201** |
| Real-IAD | 0.1732 | 0.1732 | 0.2171 | 0.2288 | 5.9602 | 15.567 | 16.051 | **0.0192** |

*Table 3.* Ablation studies on MVTec-AD: Evaluating the effects of fixed/adaptive masking strategies and distillation methods on anomaly detection/localization using AU-ROC. Abbreviations: F-Mask (Fixed Mask), A-Mask (Adaptive Mask), ADD(Adversarial Diffusion Distillation), Distill (Score Distillation), Adv. D. (Adversarial Score Distillation).

| DDPM | F-Mask | A-Mask | ADD | Distill | Adv. D. | Results |
|---|---|---|---|---|---|---|
| ✓ | - | - | - | - | - | 90.2 / 91.4 |
| ✓ | ✓ | - | - | - | - | 97.7 / 96.8 |
| ✓ | - | ✓ | - | - | - | 98.4 / 97.5 |
| ✓ | - | ✓ | ✓ | - | - | 94.5 / 95.2 |
| ✓ | - | ✓ | - | ✓ | - | 98.3 / 97.2 |
| ✓ | - | ✓ | - | ✓ | ✓ | **98.8 / 97.7** |

datasets due to its one-step diffusion process. On MVTec-AD, OmiAD records an inference time of 0.0254s, dramatically outperforming DDAD (6.0067s), DiAD (15.662s), and TransFusion (15.904s), achieving a nearly 200× speedup over DDAD. Similarly, on VisA, MPDD, and Real-IAD, OmiAD maintains its efficiency, with inference times of 0.0283s, 0.0201s, and 0.0192s, respectively. Compared to Transformer-based methods, OmiAD delivers a 7× to 8× speedup, significantly surpassing HVQ-Trans (0.1795s) and UniAD (0.1891s). Additionally, it outperforms Reconstruction-based methods such as RD4AD (0.2075s) and ReContrast (0.2383s). These results underscore OmiAD's ability to balance accuracy and efficiency, making it a highly practical solution for real-time anomaly detection.

### 4.5. Ablation Studies and Analysis

Table 3 presents the ablation study results on the MVTec-AD dataset, examining the impact of masking strategies and distillation techniques on anomaly detection and localization performance. The results, reported using the AU-ROC metric for both image-level detection and pixel-level localization, also include a comparison between our proposed ASD and Adversarial Diffusion Distillation (ADD) (Sauer et al., 2025), highlighting the effectiveness of our distillation strategy.

**Impact of Masking Strategies:** The introduction of masking strategies significantly enhances model performance by reducing shortcut learning. Specifically, the Adaptive

Masking (A-Mask) strategy demonstrates superior performance compared to the Fixed Masking (F-Mask) approach, demonstrating that dynamically adjusting masking patterns based on input characteristics enables the model to better capture global context. This, in turn, improves its ability to reconstruct anomalies as their normal counterparts, leading to superior detection performance.

**Effect of Distillation and Adversarial Distillation:** Introducing Score Distillation (Distill) allows the student model to closely approximate the performance of the multi-step teacher model while drastically reducing inference time. Furthermore, integrating Adversarial Score Distillation (Adv. D.) further enhances model effectiveness, even surpassing the performance of the teacher model. This improvement is attributed to the shared-weight discriminator embedded within the student network, which refines the alignment between generated and real samples, ensuring better adaptation to the normal data distribution. Notably, the highest AU-ROC scores of 98.8/97.7 are achieved when combining Adaptive Masking with Adversarial Score Distillation.

**Comparison with Adversarial Diffusion Distillation:** To further evaluate the effectiveness of our proposed ASD, developed based on SiD (Zhou et al., 2024b) and SiDA (Zhou et al., 2024a), we conducted a direct comparison with ADD (Sauer et al., 2025). ASD achieves a substantial performance gain, attaining 98.8/97.7 AU-ROC, significantly outperforming ADD, which achieves 94.5/95.2 AU-ROC. These results highlight the superior robustness of ASD in handling diverse anomaly patterns, further demonstrating its effectiveness in improving anomaly detection and localization accuracy.

**Overall Impact on Performance:** The integration of adaptive masking strategies and adversarial score distillation underscores the critical role of mitigating shortcut learning while enhancing the efficiency of the diffusion process. By facilitating a deeper understanding of global context and ensuring accurate reconstructions, OmiAD achieves state-of-the-art performance across a variety of anomaly detection datasets. For further details and additional ablation studies, please refer to Appendix I.

# 5. Conclusion

In this paper, we proposed OmiAD, a novel one-step Adaptive Masked Diffusion Models for multi-class anomaly detection, addressing two key challenges: shortcut learning and inference efficiency. To reduce shortcut reliance, we introduced an Adaptive Masking strategy, guiding the model to leverage global contextual information and reconstruct anomalies into their normal counterparts, thereby improving robustness across diverse scenarios. Furthermore, we tackled the efficiency bottleneck of diffusion models by designing an adversarial score distillation framework. Experiments on four datasets show that OmiAD achieves state-of-the-art performance across seven metrics, offering a practical solution for real-time industrial anomaly detection.

# Acknowledgements

The work of Y. Feng, W. Chen, Y. Li, B. Chen, Y. Wang, Z. Zhao, and H. Liu was supported in part by the National Natural Science Foundation of China under Grant U21B2006; in part by the Fundamental Research Funds for the Central Universities QTZX24003 and QTZX23018; in part by the 111 Project under Grant B18039; and in part by Shaanxi Youth Innovation Team Project. W. Chen acknowledges the support of the stabilization support of National Radar Signal Processing Laboratory under Grant (JKW202X0X, KGJ202401) and National Natural Science Foundation of China (NSFC) (6220010437).

# Impact Statement

This paper presents work whose goal is to advance the field of Machine Learning. There are many potential societal consequences of our work, none which we feel must be specifically highlighted here.

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

## A. Algorithm for Training OmiAD

---
**Algorithm 1** Training for OmiAD

---
1: **Input:** One step Generator $g_\theta$, pretrained teacher network $\epsilon_\phi$, student score network $\epsilon_\psi$, $t_{\text{init}}$, $t_{\max} = 990$, $\alpha = 1$, $\lambda_\theta^{\text{adv}} = 1$, $\lambda_\psi^{\text{adv}} = 0.01$, latent discriminator map size $(W', H')$.
2: **Initialization:** $\theta \leftarrow \phi, \psi \leftarrow \phi, D(\cdot) \leftarrow \text{encoder}(\psi)$
3: **repeat**
4:    Sample $x_0$ from the original dataset.
5:    Compute $x_g = g_\theta\left(\sqrt{\bar{\alpha}_{t_{\text{init}}}} x_0 + \sqrt{1 - \bar{\alpha}_{t_{\text{init}}}} \varepsilon\right)$, where $\varepsilon \sim \mathcal{N}(0, \mathbf{I})$.
6:    Sample $t \sim \text{Uniform}(0, \ldots, t_{\max})$, $\epsilon_t \sim \mathcal{N}(0, \mathbf{I})$, and compute $x_t = \sqrt{\bar{\alpha}_t} x_g + \sqrt{1 - \bar{\alpha}_t} \epsilon_t$, $y_t = \sqrt{\bar{\alpha}_t} x_0 + \sqrt{1 - \bar{\alpha}_t} \epsilon_t$.
7:    Update $\psi$ with the loss function:
8:       $L_\psi^{\text{adv}} = \frac{1}{W'H'} \sum_{i'=1}^{W'} \sum_{j'=1}^{H'} [\ln D(y_t)[i', j'] + \ln(1 - D(x_t)[i', j'])]$
9:       $\mathcal{L}_\psi = \gamma(t)\left(\|\epsilon_\psi(x_t, t) - \epsilon_t\|_2^2 + \lambda_\psi^{\text{adv}} L_\psi^{\text{adv}}\right)$
10:      $\psi = \psi - \eta\nabla_\psi \mathcal{L}_\psi$
11:   Sample $t \sim \text{Uniform}(0, \ldots, t_{\max})$, $\epsilon_t \sim \mathcal{N}(0, \mathbf{I})$, and compute $x_t = \sqrt{\bar{\alpha}_t} x_g + \sqrt{1 - \bar{\alpha}_t} \epsilon_t$.
12:   Update $g_\theta$ with the loss function:
13:      $\mathcal{L}_\theta^{\text{adv}} = \frac{1}{W'H'} \sum_{i=1}^{W'} \sum_{j=1}^{H'} D(x_t)[i, j]$
14:      $\mathcal{L}_\theta = \gamma(t)(\epsilon_\phi(x_t, t) - \epsilon_\psi(x_t, t))(\epsilon_\psi(x_t, t) - \epsilon_t) - \alpha\gamma(t)\|\epsilon_\phi(x_t, t) - \epsilon_\psi(x_t, t)\|_2^2 + \lambda_\theta^{\text{adv}} L_\theta^{\text{adv}}$
15:      $\theta = \theta - \eta\nabla_\theta \mathcal{L}_\theta$
16: **until** convergence
17: **Output:** $g_\theta$

---

## B. Algorithm for Inference with OmiAD

---
**Algorithm 2** OmiAD Inference Stage

---
1: **Input:** $img$: Original input image, $g_\theta$: Trained one-step generator, $EfficientNet$: Feature extractor, $t_{init}$.
2: **Output:**
3: $S$: Pixel-wise anomaly score map (same resolution as input image)
4: **Procedure:**
5:    Feature Extraction: $x_0 = EfficientNet(img)$
6:    Noising: $x_t = \sqrt{\bar{\alpha}_t} \cdot x_0 + \sqrt{1 - \bar{\alpha}_t} \cdot \varepsilon$, $t = t_{init}$, $\varepsilon \sim \mathcal{N}(0, I)$,
7:    One-step Reconstruction: $\hat{x}_0 = g_\theta(x_t)$
8:    Anomaly Score Computation: $S = \|x_0 - \hat{x}_0\|_2^2$

---

## C. Derivation of the Optimized Distillation Loss Function

Following Zhou et al. (2024b; 2025), we start with the original Fisher divergence-based loss function:

$$\mathcal{L}_\theta = \mathbb{E}_{q(x_t|x_g, t)\, p_\theta(x_g)} \left[\gamma(t) \left\|\epsilon_{\psi^*(\theta)}(x_t, t) - \epsilon_\phi(x_t, t)\right\|_2^2\right] \tag{19}$$

where $\epsilon_\phi$ and $\epsilon_{\psi^*(\theta)}$ represent the teacher and optimal student scores, respectively; $\gamma(t)$ is a weighting function dependent on $t$; $q(x_t \mid x_g, t)$ describes the noisy data distribution conditioned on $x_g$ and $t$; and $p_\theta(x_g)$ reflects the data distribution tied to $\theta$.

We refer to a naive approximation of this loss—obtained by substituting $\psi^*(\theta)$ with its approximation $\psi$—as $\mathcal{L}_\theta^{(1)}$, expressed as

$$\mathcal{L}_\theta^{(1)} = \mathbb{E}_{q(x_t|x_g, t)\, p_\theta(x_g)} \left[\gamma(t) \left\|\epsilon_\psi(x_t, t) - \epsilon_\phi(x_t, t)\right\|_2^2\right] \tag{20}$$

Using the definition of the student score

$$\epsilon_{\psi^*(\theta)}(x_t, t) = -\sqrt{1 - \bar{\alpha}_t} \nabla_{x_t} \log p_\theta(x_t) \tag{21}$$

the objective function in Equation 19 can be equivalently reformulated as follows:

$$
\begin{aligned}
&\mathbb{E}_{q(x_t|x_g,t)\,p_\theta(x_g)}\Big[\big\|\boldsymbol{\epsilon}_\phi(\boldsymbol{x_t},t) - \boldsymbol{\epsilon}_{\psi^*(\theta)}(\boldsymbol{x_t},t)\big\|^2\Big]\\
&= \mathbb{E}_{q(x_t|x_g,t)\,p_\theta(x_g)}\Big[\big\langle \boldsymbol{\epsilon}_\phi(\boldsymbol{x_t},t) - \boldsymbol{\epsilon}_{\psi^*(\theta)}(\boldsymbol{x_t},t), \boldsymbol{\epsilon}_\phi(\boldsymbol{x_t},t) - \boldsymbol{\epsilon}_{\psi^*(\theta)}(\boldsymbol{x_t},t)\big\rangle\Big]\\
&= \mathbb{E}_{q(x_t|x_g,t)\,p_\theta(x_g)}\Big[\big\langle \boldsymbol{\epsilon}_\phi(\boldsymbol{x_t},t) - \boldsymbol{\epsilon}_{\psi^*(\theta)}(\boldsymbol{x_t},t), \boldsymbol{\epsilon}_\phi(\boldsymbol{x_t},t)\big\rangle\Big]\\
&\quad + \mathbb{E}_{q(x_t|x_g,t)\,p_\theta(x_g)}\Big[\big\langle \boldsymbol{\epsilon}_\phi(\boldsymbol{x_t},t) - \boldsymbol{\epsilon}_{\psi^*(\theta)}(\boldsymbol{x_t},t), \sqrt{1-\bar\alpha_t}\nabla_{x_t}\log p_\theta(\boldsymbol{x_t})\big\rangle\Big]
\end{aligned}
\tag{22}
$$

Furthermore, by using a score-related identity given by

$$
\nabla_{x_t}\log p(\boldsymbol{x_t}) = \mathbb{E}_{x_0\sim p(x_0|x_t)}\big[\nabla_{x_t}\log p(\boldsymbol{x_t}\mid\boldsymbol{x_0})\big]
\tag{23}
$$

and more specifically the score-projection identity discussed in SiD, we obtain

$$
\begin{aligned}
&\mathbb{E}_{q(x_t|x_g,t)\,p_\theta(x_g)}\Big[\big\langle \boldsymbol{\epsilon}_\phi(\boldsymbol{x_t},t) - \boldsymbol{\epsilon}_{\psi^*(\theta)}(\boldsymbol{x_t},t), \sqrt{1-\bar\alpha_t}\nabla_{x_t}\log p_\theta(\boldsymbol{x_t})\big\rangle\Big]\\
&= \mathbb{E}_{p(x_g|x_t)p_\theta(x_t)}\Big[\big\langle \boldsymbol{\epsilon}_\phi(\boldsymbol{x_t},t) - \boldsymbol{\epsilon}_{\psi^*(\theta)}(\boldsymbol{x_t},t), \sqrt{1-\bar\alpha_t}\nabla_{x_t}\log p(\boldsymbol{x_t}\mid\boldsymbol{x_g})\big\rangle\Big]\\
&= -\mathbb{E}_{q(x_t|x_g,t)p_\theta(x_g)}\Big[\big\langle \boldsymbol{\epsilon}_\phi(\boldsymbol{x_t},t) - \boldsymbol{\epsilon}_{\psi^*(\theta)}(\boldsymbol{x_t},t), \frac{\boldsymbol{x_t} - \sqrt{\bar\alpha_t}\boldsymbol{x_g}}{\sqrt{1-\bar\alpha_t}}\big\rangle\Big]\\
&= -\mathbb{E}_{x_g\sim p_\theta(x_g),\epsilon\sim\mathcal{N}(0,I)}\Big[\big\langle \boldsymbol{\epsilon}_\phi(\boldsymbol{x_t},t) - \boldsymbol{\epsilon}_{\psi^*(\theta)}(\boldsymbol{x_t},t), \boldsymbol{\epsilon}\big\rangle\Big]
\end{aligned}
\tag{24}
$$

Plugging Equation 24 into Equation 22, we obtain an equivalent expression of the objective function in Equation 19 as

$$
\mathcal{L}_\theta = \mathbb{E}_{q(x_t|x_g,t)\,p_\theta(x_0)}\Big[\gamma(t)\,(\boldsymbol{\epsilon}_\phi(\boldsymbol{x_t},t) - \boldsymbol{\epsilon}_{\psi^*(\theta)}(\boldsymbol{x_t},t))(\boldsymbol{\epsilon}_\phi(\boldsymbol{x_t},t) - \boldsymbol{\epsilon})\Big]
\tag{25}
$$

Substituting $\psi^*(\theta)$ with its approximation $\psi$, we obtain another approximated Fisher divergence as

$$
\mathcal{L}_\theta^{(2)} = \mathbb{E}_{q(x_t|x_g,t)\,p_\theta(x_0)}\Big[\gamma(t)\,(\boldsymbol{\epsilon}_\phi(\boldsymbol{x_t},t) - \boldsymbol{\epsilon}_\psi(\boldsymbol{x_t},t))(\boldsymbol{\epsilon}_\phi(\boldsymbol{x_t},t) - \boldsymbol{\epsilon})\Big]
\tag{26}
$$

Compared with $\mathcal{L}_\theta^{(1)}$ in Equation 20, the new loss $\mathcal{L}_\theta^{(2)}$ leverages the score-projection identity to mitigate the intricate dependency of $\psi^*(\theta)$ on $\theta$.

The final loss, designed by SiD to counteract the effect of ignoring the dependency of $\psi^*(\theta)$ on $\theta$, is defined as

$$
\begin{aligned}
\mathcal{L}_\theta &= \mathcal{L}_\theta^{(2)} - \alpha\mathcal{L}_\theta^{(1)}\\
&= \mathbb{E}_{q(x_t|x_g,t)p_\theta(x_g)}\Big[\gamma(t)\;\Big(\boldsymbol{\epsilon}_\psi(\boldsymbol{x_t},t) - \boldsymbol{\epsilon}_\phi(\boldsymbol{x_t},t)(\boldsymbol{\epsilon} - \boldsymbol{\epsilon}_\phi(\boldsymbol{x_t},t)) - \alpha\|\boldsymbol{\epsilon}_\psi(\boldsymbol{x_t},t) - \boldsymbol{\epsilon}_\phi(\boldsymbol{x_t},t)\|_2^2\Big)\Big]
\end{aligned}
\tag{27}
$$

## D. Dataset:

**MVTec-AD dataset:** The MVTec Anomaly Detection (MVTec-AD) dataset (Bergmann et al., 2019) is a widely used benchmark for unsupervised anomaly detection and localization in industrial scenarios. It includes 15 categories of high-resolution images across both texture and object types, covering a variety of real-world defects such as scratches, dents, and cracks. The dataset consists of 3,629 normal images for training and 1,725 test images containing both normal and anomalous samples. Additionally, pixel-level ground truth annotations are provided for anomaly localization, making it suitable for evaluating both detection accuracy and localization performance.

**VisA dataset:** The VisA dataset is a large-scale anomaly detection benchmark (Zou et al., 2022) that features 10,821 high-resolution images across 12 distinct object categories. Each category contains various types of anomalies, ranging from structural defects to surface irregularities. The dataset is structured into three types of anomaly scenarios: complex structures, multiple instances, and single-instance anomalies. With detailed annotations and a diverse set of defect types, VisA is particularly useful for evaluating the robustness of anomaly detection models under different industrial conditions.

**MPDD dataset:** The MPDD dataset (Jezek et al., 2021) focuses on anomaly detection in metallic components during the manufacturing process. It contains six categories of painted metal parts with various types of defects, such as scratches, dents, and discoloration. The dataset provides 888 normal training images and 458 test images, comprising both normal and defective samples. Unlike other datasets, MPDD introduces challenges such as varying spatial orientations, lighting conditions, and non-homogeneous backgrounds, making it a realistic and challenging benchmark for defect detection.

**Real-IAD dataset:** The Real-IAD dataset (Wang et al., 2024) is one of the largest publicly available datasets for industrial anomaly detection, consisting of over 150,000 high-resolution images across 30 object categories. The dataset includes 99,721 normal images and 51,329 images with various types of anomalies. Real-IAD covers a wide range of real-world industrial defects and includes complex, multi-instance, and subtle anomalies, providing a comprehensive benchmark for evaluating anomaly detection methods in practical applications.

## E. Baselines:

We conduct and analyze a variety of qualitative and quantitative comparison experiments on MVTec-AD, VisA, MPDD, and Real-IAD. For the comparison, we select three embedding-based methods, DeSTSeg (Zhang et al., 2023b), RD4AD (Deng & Li, 2022), and SimpleNet (Liu et al., 2023), along with one reconstruction-based method, UniAD (You et al., 2022). Additionally, we include the unified SOTA method HVQ-Trans (Lu et al., 2023) and the diffusion-based method DiAD (He et al., 2024b).

Furthermore, we select ReContrast (Guo et al., 2023), DADD (Mousakhan et al., 2023) and TransFusion (Fučka et al., 2024) to evaluate inference speed.

## F. Implementation details:

The input image resolution is set to $224 \times 224$, and the feature maps are resized to $32 \times 32$. Feature maps from stages 1 to 4 of EfficientNet-b4 (Tan & Le, 2019) are resized and concatenated to produce a 272-channel feature representation. For the diffusion model, we configure the number of timesteps as $T = 1000$, employing a linear noise schedule with $\beta_1 = 10^{-4}$ and $\beta_T = 0.02$, following the setup in Ho et al. (2020).In our adaptive masking strategy, the mask probability is controlled by the following parameters: $p_{\min} = 0.1$, $p_{\max} = 0.4$, and $k = 2$. The model is trained using the Adam optimizer (Kingma, 2014) with a learning rate of 0.001 and a batch size of 32. All experiments are implemented in PyTorch 2.1.0 (Paszke et al., 2019) and executed on an NVIDIA RTX 4090 GPU with 24GB of VRAM.

## G. More Quantitative Results for Each Category on MVTec-AD, VisA, MPDD, and Real-IAD:

To provide a more comprehensive evaluation, we present additional quantitative results for each category across the four benchmark datasets: MVTec-AD, VisA, MPDD, and Real-IAD. The reported evaluation metrics include the Area Under the Receiver Operating Characteristic Curve (AU-ROC), Average Precision (AP) , and F1-score-max (F1_max). Additionally, for anomaly segmentation, we report the Area Under the Per-Region-Overlap (AU-PRO) .

These detailed per-category results provide a deeper understanding of the model's performance across different types of anomalies. By analyzing the results on each dataset, we can better observe the robustness and adaptability of the proposed method in various real-world scenarios, demonstrating its effectiveness in both anomaly detection and segmentation tasks.

*Table 4.* Comparison with SoTA methods on MVTec-AD dataset for multi-class anomaly detection with AU-ROC/AP/F1_max metrics.

| | Category | RD4AD | UniAD | SimpleNet | DeSTSeg | DiAD | HVQ-Trans | Ours |
|---|---|---|---|---|---|---|---|---|
| Object | Bottle | 99.6/99.9/98.4 | 99.7/**100./100.** | **100./100./100.** | 98.7/99.6/96.8 | 99.7/96.5/91.8 | **100./100./100.** | **100./100./100.** |
| | Cable | 84.1/89.5/82.5 | 95.2/95.9/88.0 | 97.5/98.5/94.7 | 89.5/94.6/85.9 | 94.8/98.8/95.2 | **99.0**/98.8/95.1 | 98.4/**99.4/95.6** |
| | Capsule | 94.1/96.9/**96.9** | 86.9/97.8/94.4 | 90.7/97.9/93.5 | 82.8/95.9/92.6 | 89.0/97.5/95.5 | **95.4**/99.2/96.3 | 94.7/**99.3**/96.8 |
| | Hazelnut | 60.8/69.8/86.4 | 99.8/**100.**/99.3 | 99.9/99.9/99.3 | 98.8/99.2/98.6 | 99.5/99.7/97.3 | **100./100.**/99.3 | **100./100./100.** |
| | MetalNut | **100./100.**/99.5 | 99.2/99.9/**99.5** | 96.9/99.3/96.1 | 92.9/98.4/92.2 | 99.1/96.0/91.6 | 99.9/99.9/98.9 | 99.4/99.9/98.9 |
| | Pill | **97.5/99.6/96.8** | 93.7/98.7/95.7 | 88.2/97.7/92.5 | 77.1/94.4/91.7 | 95.7/98.5/94.5 | 95.8/99.2/94.9 | 94.2/99.2/95.4 |
| | Screw | **97.7**/99.3/95.8 | 87.5/96.5/89.0 | 76.7/90.6/87.7 | 69.9/88.4/85.4 | 90.7/**99.7/97.9** | 95.6/97.9/92.1 | 96.9/98.8/96.3 |
| | Toothbrush | 97.2/99.0/94.7 | 94.2/97.4/95.2 | 89.7/95.7/92.3 | 71.7/89.3/84.5 | **99.7**/99.9/99.2 | 93.6/99.9/98.4 | **99.7/100./100.** |
| | Transistor | 94.2/95.2/90.0 | 99.8/98.0/93.8 | 99.2/98.7/97.6 | 78.2/79.5/68.8 | 99.8/99.6/97.4 | 99.7/99.5/96.4 | **99.9/99.9/98.8** |
| | Zipper | 99.5/99.9/99.2 | 95.8/99.5/97.1 | 99.0/99.7/98.3 | 88.4/96.3/93.1 | 95.1/99.1/94.4 | 97.9/99.6/98.3 | **99.8/100./99.6** |
| Texture | Carpet | 98.5/99.6/97.2 | 99.8/99.9/99.4 | 95.7/98.7/93.2 | 95.9/98.8/94.9 | 99.4/99.9/98.3 | **99.9/100./100.** | 99.6/**100.**/99.4 |
| | Grid | 98.0/99.4/96.5 | 98.2/99.5/97.3 | 97.6/99.2/96.4 | 97.9/99.2/96.6 | 98.5/99.8/97.7 | 97.0/99.5/97.3 | **99.8/99.9/99.1** |
| | Leather | **100./100./100.** | **100./100./100.** | **100./100./100.** | 99.2/99.8/98.9 | 99.8/99.7/97.6 | **100./100./100.** | **100./100./100.** |
| | Tile | 98.3/99.3/96.4 | 99.3/99.8/98.2 | 99.3/99.8/**98.8** | 97.0/98.9/95.3 | 96.8/**99.9**/98.4 | 99.2/99.8/98.2 | **100.**/99.9/98.8 |
| | Wood | 99.2/99.8/98.3 | 98.6/99.6/96.6 | 98.4/99.5/96.7 | **99.9/100.**/99.2 | 99.7/**100./100.** | 97.2/99.6/97.4 | 99.0/99.8/98.3 |
| | Mean | 94.6/96.5/95.2 | 96.5/98.8/96.2 | 95.3/98.4/95.8 | 89.2/95.5/91.6 | 97.2/99.0/96.5 | 98.0/99.5/97.5 | **98.8/99.7/98.5** |

*Table 5.* Comparison with SoTA methods on MVTec-AD dataset for multi-class anomaly localization with AU-ROC/AP/F1_max metrics/AU-PRO metrics.

| | Category | RD4AD | UniAD | SimpleNet | DeSTSeg | DiAD | HVQ-Trans | Ours |
|---|---|---|---|---|---|---|---|---|
| Object | Bottle | 97.8/68.2/67.6/94.0 | 98.1/66.0/69.2/93.1 | 97.2/53.8/62.4/89.0 | 93.3/61.7/56.0/67.5 | 98.4/52.2/54.8/86.6 | 98.3/71.8/70.2/94.6 | **98.6/74.9/73.8/95.6** |
| | Cable | 85.1/26.3/33.6/75.1 | 97.3/39.9/45.2/86.1 | 96.7/42.4/51.2/85.4 | 89.3/37.5/40.5/49.4 | 96.8/50.1/57.8/80.5 | 98.1/52.4/59.0/87.5 | **98.3/60.5/62.9/91.8** |
| | Capsule | 98.8/43.4/50.0/**94.8** | 98.5/42.7/46.5/92.1 | 98.5/05.4/44.3/84.5 | 95.8/47.9/48.9/62.1 | 97.1/42.0/45.3/87.2 | 98.8/45.3/49.7/90.7 | **98.9/48.1/52.0**/92.5 |
| | Hazelnut | 97.9/36.2/51.6/92.7 | 98.1/55.2/56.8/94.1 | 98.4/44.6/51.4/87.4 | 98.2/65.8/61.6/84.5 | 98.3/**79.2/80.4**/91.5 | 96.3/67.6/63.2/92.5 | 98.6/59.6/59.9/**94.3** |
| | MetalNut | 94.8/55.5/66.4/**91.9** | 62.7/14.6/29.2/81.8 | **98.0/83.1/79.4**/85.2 | 84.2/42.0/22.8/53.0 | 97.3/30.0/38.3/90.6 | 96.3/67.1/75.5/90.9 | 96.5/66.6/75.6/90.3 |
| | Pill | **97.5**/63.4/65.2/95.8 | 95.0/44.0/53.9/95.3 | 96.5/**72.4/67.7**/81.9 | 96.2/61.7/41.8/27.9 | 95.7/46.0/51.4/89.0 | 97.1/50.1/57.6/94.9 | 96.6/56.8/60.7/**95.9** |
| | Screw | 99.4/40.2/44.6/96.8 | 98.3/28.7/37.6/95.2 | 96.5/15.9/23.2/84.0 | 93.8/19.9/25.3/47.3 | 97.9/**60.6/59.6**/95.0 | 98.9/28.8/36.2/94.3 | **99.5**/38.7/43.5/**97.2** |
| | Toothbrush | **99.0**/53.6/58.8/92.0 | 98.4/34.9/45.7/87.9 | 98.4/46.9/52.5/87.4 | 96.2/52.9/58.8/30.9 | **99.0**/78.7/72.8/95.0 | 98.6/40.8/51.4/89.2 | 98.7/40.5/56.2/91.1 |
| | Transistor | 85.9/42.3/45.2/74.7 | 97.9/59.5/64.6/93.5 | 95.8/58.2/56.0/83.2 | 73.6/38.4/39.2/43.9 | 95.1/15.6/31.7/90.0 | 97.9/71.2/67.2/95.4 | **98.4/73.4/72.5/96.1** |
| | Zipper | 98.5/53.9/**60.3**/94.1 | 97.9/53.4/54.6/90.7 | 97.9/53.4/54.6/90.7 | 97.3/**64.7**/59.2/66.9 | 96.2/60.7/60.0/91.6 | 97.5/38.7/48.8/91.7 | **98.6**/52.7/59.3/**95.6** |
| Texture | Carpet | **99.0**/58.5/**60.4**/95.1 | 98.5/49.9/51.1/94.4 | 97.4/38.7/43.2/90.6 | 93.6/**59.9**/58.9/89.3 | 98.6/42.2/46.4/90.6 | 98.7/57.5/57.7/94.7 | 98.5/59.2/54.8/94.6 |
| | Grid | 96.5/23.0/28.4/**97.0** | 63.1/00.7/01.9/92.9 | 96.8/20.5/27.6/88.6 | 97.0/42.1/46.9/86.8 | 96.6/**66.0/64.1**/94.0 | 97.0/24.5/30.5/89.5 | **98.5**/35.4/37.1/95.5 |
| | Leather | 99.3/38.0/45.1/97.4 | 98.8/32.9/34.4/96.8 | 98.7/28.5/32.9/92.7 | **99.5/71.5/66.5**/91.1 | 98.8/56.1/62.3/91.3 | 98.8/33.7/36.6/**97.6** | 98.9/36.3/39.4/96.9 |
| | Tile | 95.3/48.5/60.5/85.8 | 91.8/42.1/50.6/78.4 | **95.7**/60.5/59.9/90.6 | 93.0/**71.0/66.2**/87.1 | 92.4/65.7/64.1/**90.7** | 92.2/41.6/52.9/81.2 | 92.7/47.5/54.7/82.2 |
| | Wood | 95.3/47.8/51.0/90.0 | 93.2/37.2/41.5/86.7 | 91.4/34.8/39.7/76.3 | **95.9/77.3/71.3**/83.4 | 93.3/43.3/43.5/**97.5** | 92.4/37.2/42.6/86.6 | 94.2/44.5/48.0/88.5 |
| | Mean | 96.1/48.6/53.8/91.1 | 96.8/43.4/49.5/90.7 | 96.9/45.9/49.7/86.5 | 93.1/**54.3**/50.9/64.8 | 96.8/52.6/55.5/90.7 | 97.3/48.2/53.3/91.4 | **97.7**/52.6/**56.7/93.2** |

*Table 6.* Comparison with SoTA methods on VisA dataset for multi-class anomaly detection with AU-ROC/AP/F1_max metrics.

| | Category | RD4AD | UniAD | SimpleNet | DeSTSeg | DiAD | HVQ-Trans | Ours |
|---|---|---|---|---|---|---|---|---|
| Complex structure | PCB1 | 96.2/95.5/91.9 | 92.8/92.7/87.8 | 91.6/91.9/86.0 | 87.6/83.1/83.7 | 88.1/88.7/80.7 | 96.7/93.2/87.7 | **97.8/97.4/95.1** |
| | PCB2 | **97.8**/97.8/**94.2** | 87.8/87.7/83.1 | 92.4/93.3/84.5 | 86.5/85.8/82.6 | 91.4/91.4/84.7 | 93.4/94.8/88.0 | **97.8/98.5**/94.1 |
| | PCB3 | 96.4/**96.2**/91.0 | 78.6/78.6/76.1 | 89.1/91.1/82.6 | 93.7/95.1/87.0 | 86.2/87.6/77.6 | 92.0/87.1/79.5 | **96.7**/95.1/87.6 |
| | PCB4 | 99.9/99.9/99.0 | 98.8/98.8/94.3 | 97.0/97.0/93.5 | 97.8/97.8/92.7 | 99.6/99.5/97.0 | 99.5/99.0/97.0 | **100./100./99.0** |
| Multiple instances | Macaroni 1 | 75.9/61.5/76.8 | 79.9/79.8/72.7 | 85.9/82.5/73.1 | 76.6/69.0/71.0 | 85.7/85.2/78.8 | 93.1/84.1/79.8 | **97.3/97.5/92.8** |
| | Macaroni 2 | **88.3/84.5/83.8** | 71.6/71.6/69.9 | 68.3/54.3/59.7 | 68.9/62.1/67.7 | 62.5/57.4/69.6 | 86.2/84.1/81.5 | 85.1/83.3/79.5 |
| | Capsules | 82.2/90.4/81.3 | 55.6/55.6/76.9 | 74.1/82.8/74.6 | **87.1/93.0/84.2** | 58.2/69.0/78.5 | 77.1/83.4/77.5 | 85.7/89.0/78.8 |
| | Candles | 92.3/92.9/86.0 | 94.1/94.0/86.1 | 84.1/73.3/76.6 | 94.9/94.8/89.2 | 92.8/92.0/87.6 | 96.8/98.0/93.1 | **97.4/98.6/93.4** |
| Single instance | Cashew | 92.0/95.8/90.7 | 92.8/92.8/**91.4** | 88.0/91.3/84.7 | 92.0/96.1/88.1 | 91.5/95.7/89.7 | **94.9/96.8**/90.4 | 93.3/96.4/90.9 |
| | Chewing gum | 94.9/97.5/92.1 | 96.3/96.2/95.2 | 96.4/98.2/93.8 | 95.8/98.3/94.7 | 99.1/99.5/95.9 | **99.4/99.6/97.5** | 99.2/**99.8**/97.5 |
| | Fryum | **95.3/97.9**/91.5 | 83.0/83.0/85.0 | 88.4/93.0/83.3 | 92.1/96.1/89.5 | 89.8/95.0/87.2 | 90.4/94.5/84.9 | 94.0/96.5/88.5 |
| | Pipe fryum | 97.9/98.9/96.5 | 94.7/94.7/93.9 | 90.8/95.5/88.6 | 94.1/97.1/91.9 | 96.2/98.1/93.7 | 98.5/98.4/94.0 | **98.9/99.4/97.0** |
| | Mean | 92.4/92.4/89.6 | 85.5/85.5/84.4 | 87.2/87.0/81.8 | 88.9/89.0/85.2 | 86.8/88.3/85.1 | 93.2/92.8/87.6 | **95.3/96.0/91.2** |

*Table 7.* Comparison with SoTA methods on VisA dataset for multi-class anomaly localization with AU-ROC/AP/F1_max/AU-PRO metrics.

| Category | | RD4AD | UniAD | SimpleNet | DeSTSeg | DiAD | HVQ-Trans | Ours |
|---|---|---|---|---|---|---|---|---|
| Complex structure | PCB1 | 99.4/66.2/62.4/**95.8** | 93.3/03.9/08.3/64.1 | 99.2/**86.1/78.8**/83.6 | 95.8/46.4/49.0/83.2 | 98.7/49.6/52.8/80.2 | 99.4/63.1/58.8/87.4 | **99.7**/70.8/66.2/93.0 |
| | PCB2 | 98.0/**22.3/30.0**/90.8 | 93.9/04.2/09.2/66.9 | 96.6/08.9/18.6/85.7 | 97.3/14.6/28.2/79.9 | 95.2/07.5/16.7/67.0 | 98.0/10.1/18.2/82.0 | **98.9**/16.6/22.2/87.1 |
| | PCB3 | 97.9/26.2/35.2/**93.9** | 97.3/13.8/21.9/70.6 | 97.2/**31.0/36.1**/85.1 | 97.7/28.1/33.4/62.4 | 96.7/08.0/18.8/68.9 | 98.3/21.1/23.7/80.5 | **99.1**/29.7/30.4/87.2 |
| | PCB4 | 97.8/31.4/37.0/**88.7** | 94.9/14.7/22.9/72.3 | 93.9/23.9/32.9/61.1 | 95.8/**53.0/53.2**/76.9 | 97.0/17.6/27.2/85.0 | 97.7/21.1/29.8/85.9 | **98.1**/42.0/44.1/86.4 |
| Multiple instances | Macaroni 1 | 99.4/02.9/06.9/95.3 | 97.4/03.7/09.7/84.0 | 98.9/03.5/08.4/92.0 | 99.1/05.8/13.4/62.4 | 94.1/10.2/16.7/68.5 | 99.4/09.9/19.3/91.2 | **99.7**/20.1/**29.8/96.3** |
| | Macaroni 2 | **99.7**/13.2/21.8/**97.4** | 95.2/00.9/04.3/76.6 | 93.2/00.6/03.9/77.8 | 98.5/06.3/14.4/70.0 | 93.6/00.90/2.8/73.1 | 98.5/05.5/13.8/91.3 | 99.4/08.5/15.4/94.0 |
| | Capsules | **99.4**/60.4/**60.8**/93.1 | 88.7/03.0/07.4/43.7 | 97.1/52.9/53.3/73.7 | 96.9/33.2/09.1/76.7 | 97.3/10.0/21.0/77.9 | 99.0/51.9/55.2/76.3 | **99.4**/62.7/59.6/90.8 |
| | candle | 99.1/25.3/35.8/94.9 | 98.5/17.6/27.9/91.6 | 97.6/08.4/16.5/87.6 | 98.7/**39.9/45.8**/69.0 | 97.3/12.8/22.8/89.4 | 99.2/20.4/30.6/92.7 | **99.4**/25.7/35.1/**97.1** |
| Single instance | Cashew | 91.7/44.2/49.7/86.2 | 98.6/51.7/58.3/87.9 | 98.9/**68.9/66.0**/84.1 | 87.9/47.6/52.1/66.3 | 90.9/53.1/60.9/61.8 | **99.2**/58.3/60.9/**89.3** | 97.2/46.3/55.3/83.8 |
| | Chewing gum | 98.7/59.9/61.7/76.9 | **98.8**/54.9/56.1/81.3 | 97.9/26.8/29.8/78.3 | **98.8**/**86.9/81.0**/68.3 | 94.7/11.9/25.8/59.5 | **98.8**/43.7/44.4/**81.7** | **98.8**/58.0/55.6/73.1 |
| | Fryum | 97.0/47.6/51.5/**93.4** | 95.9/34.0/40.6/76.2 | 93.0/39.1/45.4/85.1 | 88.1/35.2/38.5/47.7 | 97.6/**58.6/60.1**/81.3 | **97.7**/50.8/55.3/83.6 | **97.7**/47.7/55.2/87.2 |
| | Pipe fryum | 99.1/56.8/58.8/**95.4** | 98.9/50.2/57.7/91.5 | 98.5/65.6/63.4/83.0 | 98.9/**78.9/72.7**/45.9 | **99.4**/72.7/69.9/89.9 | 99.4/64.6/65.8/93.4 | 99.2/56.2/60.1/94.5 |
| Mean | | 98.1/38.0/42.6/**91.8** | 95.9/21.0/27.0/75.6 | 96.8/34.7/37.8/81.4 | 96.1/39.6/43.4/67.4 | 96.0/26.1/33.0/75.2 | 98.7/35.0/39.6/86.3 | **98.9**/**40.4/44.1**/89.2 |

*Table 8.* Comparison with SoTA methods on MPDD dataset for multi-class anomaly detection with AU-ROC/AP/F1_max metrics.

| Normal Indices | RD4AD | UniAD | SimpleNet | DeSTSeg | DiAD | HVQ-Trans | Ours |
|---|---|---|---|---|---|---|---|
| Bracket Black | 81.1/86.5/83.6 | 92.7/94.2/87.9 | 78.7/85.3/80.4 | 82.3/87.9/81.2 | 81.7/87.0/84.6 | 90.2/94.5/89.4 | **92.8/95.4/90.9** |
| Bracket Brown | 84.0/85.3/91.1 | 94.0/92.8/91.9 | 92.4/95.3/93.5 | **96.9/97.2/98.1** | 76.4/87.3/83.2 | 89.7/88.5/89.3 | 94.9/94.2/90.3 |
| Bracket White | 81.1/79.8/77.1 | 78.6/81.4/75.0 | 88.4/91.4/83.0 | **94.3/95.8/91.5** | 72.6/74.7/75.0 | 78.0/78.4/75.0 | 87.4/92.7/85.2 |
| Connector | 59.8/47.8/54.9 | 90.7/87.2/86.7 | **99.3/98.5/96.6** | 97.1/94.5/87.5 | 94.3/87.1/84.8 | 89.5/80.1/80.0 | 95.7/96.6/90.3 |
| Metal Plate | **100./100./100.** | 63.4/81.1/86.6 | **100./100./100.** | **100./100./100.** | 65.3/83.6/85.5 | 97.3/99.9/98.6 | 99.7/**100./100.** |
| Tubes | **98.6/99.5/97.8** | 73.6/85.7/82.6 | 84.6/94.1/84.6 | 87.4/94.9/85.0 | 57.5/72.6/81.7 | 74.3/87.3/82.6 | 91.4/94.1/88.7 |
| Mean | 84.1/83.2/84.1 | 82.2/87.1/85.1 | 90.6/94.1/89.7 | 93.0/95.1/90.6 | 74.6/82.1/82.5 | 86.5/88.1/85.8 | **93.7/95.5/90.9** |

*Table 9.* Comparison with SoTA methods on MPDD dataset for multi-class anomaly localization with AU-ROC/AP/F1_max/AU-PRO metrics.

| Normal Indices | RD4AD | UniAD | SimpleNet | DeSTSeg | DiAD | HVQ-Trans | Ours |
|---|---|---|---|---|---|---|---|
| Bracket Black | 97.4/**09.5**/19.0/92.1 | 94.4/00.8/01.9/82.0 | 94.9/03.6/09.1/89.3 | 95.1/**09.5/21.9**/49.0 | 91.5/01.1/02.9/74.7 | 96.1/01.1/02.7/87.3 | **98.1**/03.7/08.4/**94.8** |
| Bracket Brown | 97.3/13.0/23.4/**94.8** | 98.7/34.3/41.0/90.9 | 94.9/08.8/17.0/87.4 | 83.9/05.4/12.5/24.9 | 95.6/08.1/16.7/84.6 | 98.2/32.3/36.4/86.5 | **98.9**/42.8/44.9/93.8 |
| Bracket White | 98.3/2.6/6.4/92.2 | 94.8/00.8/03.7/76.3 | 97.8/02.2/05.6/86.4 | 95.8/04.6/12.3/63.1 | 90.8/00.5/01.6/77.4 | 94.1/00.6/02.6/79.8 | **98.3**/07.3/18.8/**90.8** |
| Connector | 97.4/12.5/20.6/90.6 | 97.6/18.6/26.1/91.4 | **98.9/56.0/55.2**/96.5 | 96.2/30.0/35.5/61.7 | 98.1/25.1/30.7/93.5 | 97.9/18.8/29.7/92.7 | **98.9**/32.3/41.2/96.3 |
| Metal Plate | **99.3/95.9/89.7**/94.5 | 93.2/50.4/63.3/79.4 | 98.2/88.9/81.3/88.1 | 98.7/94.8/88.8/74.9 | 93.8/54.9/65.7/79.6 | 96.6/73.7/74.4/86.3 | 98.5/87.4/83.8/93.4 |
| Tubes | **99.1/77.8/73.0/96.6** | 91.7/08.3/14.0/71.4 | 97.9/42.4/46.1/92.6 | 94.8/55.2/54.7/85.4 | 88.3/05.9/09.5/60.4 | 97.0/38.8/42.8/88.9 | 98.7/52.3/56.6/94.8 |
| Mean | 98.1/35.2/38.7/93.4 | 95.1/18.9/25.0/81.9 | 97.1/33.6/35.7/90.0 | 94.1/33.2/37.6/59.8 | 93.0/15.9/21.2/78.4 | 96.7/27.6/31.4/86.9 | **98.6**/37.6/42.3/**94.0** |

*Table 10.* Comparison with SoTA methods on Real-IAD dataset for multi-class anomaly detectio with AU-ROC/AP/F1_max metrics.

| Normal Indices | RD4AD | UniAD | SimpleNet | DeSTSeg | DiAD | HVQ-Trans | Ours |
|---|---|---|---|---|---|---|---|
| audiojack | 76.2/63.2/60.8 | 81.4/76.6/64.9 | 58.4/44.2/50.9 | 81.1/72.6/64.5 | 76.5/54.3/65.7 | 81.0/80.1/74.5 | **84.5/82.0/77.8** |
| bottlecap | 89.5/86.3/81.0 | 92.5/91.7/81.7 | 54.1/47.6/60.3 | 78.1/74.6/68.1 | 91.6/**94.0/87.9** | 89.0/87.6/77.3 | **93.7**/92.7/85.2 |
| buttonbattery | 73.3/78.9/76.1 | 75.9/81.6/76.3 | 52.5/60.5/72.4 | **86.7**/89.2/**83.5** | 80.5/71.3/70.6 | 82.2/88.5/78.9 | 84.9/**90.1**/80.1 |
| endcap | 79.8/84.0/77.8 | 80.9/**86.1**/78.0 | 51.6/60.8/72.9 | 77.9/81.1/77.1 | **85.1**/83.4/**84.8** | 79.7/85.2/79.4 | 79.4/80.4/80.8 |
| eraser | 90.0/88.7/79.7 | **90.3**/89.2/80.2 | 46.4/39.1/55.8 | 84.6/82.9/71.8 | 80.0/80.0/77.3 | 89.2/89.7/81.9 | 89.5/**90.2/84.2** |
| firehood | 78.3/70.1/64.5 | 80.6/74.8/66.4 | 58.1/41.9/54.4 | 81.7/72.4/67.7 | 83.3/81.7/80.5 | 93.1/85.4/83.1 | **94.1/87.6/83.3** |
| mint | 65.8/63.1/64.8 | 67.0/66.6/64.6 | 52.4/50.3/63.7 | 58.4/55.8/63.7 | **76.7**/76.7/**76.0** | 63.0/75.7/75.1 | 66.0/**77.7**/75.0 |
| mounts | 88.6/79.9/74.8 | 87.6/77.3/77.2 | 58.7/48.1/52.4 | 74.7/56.5/63.1 | 75.3/74.5/82.5 | 92.7/88.2/81.2 | **95.2/92.3/85.9** |
| pcb | 79.5/85.8/79.7 | 81.0/88.2/79.1 | 54.5/66.0/75.5 | 82.0/88.7/79.6 | 86.0/85.1/85.4 | 86.6/92.4/82.7 | **92.2/95.7/87.3** |
| phonebattery | 87.5/83.3/77.1 | 83.6/80.0/71.6 | 51.6/43.8/58.0 | 83.3/81.8/72.1 | 82.3/77.7/75.9 | 88.0/89.5/80.5 | **92.6/93.0/84.5** |
| plasticnut | 80.3/68.0/64.4 | 80.0/69.2/63.7 | 59.2/40.3/51.8 | 83.1/**75.4/66.5** | 71.9/58.2/65.6 | 76.2/57.3/53.7 | **84.2**/67.5/62.2 |
| plasticplug | 81.9/74.3/68.8 | 81.4/75.9/67.6 | 48.2/38.4/54.6 | 71.7/63.1/60.0 | 88.7/89.2/**90.9** | 92.2/92.1/84.5 | **94.1/93.2**/86.6 |
| porcelaindoll | **86.3**/76.3/71.5 | 85.1/75.2/69.3 | 66.3/54.5/52.1 | 78.7/66.2/64.3 | 72.6/66.8/65.2 | 84.7/81.5/72.9 | 86.1/**84.5/76.3** |
| regulator | 66.9/48.8/47.7 | 56.9/41.5/44.5 | 50.5/29.0/43.9 | 79.2/63.5/56.9 | 72.1/**71.4/78.2** | 69.7/27.3/37.8 | **89.5**/69.3/67.2 |
| rolledstripbase | 97.5/98.7/94.7 | 98.7/99.3/96.5 | 59.0/75.7/79.8 | 96.5/98.2/93.0 | 68.4/55.9/56.8 | 99.3/99.7/98.4 | **99.8/99.9/98.9** |
| simcardset | 91.6/91.8/84.8 | 89.7/90.3/83.2 | 63.1/69.7/70.8 | 95.5/96.2/89.2 | 72.6/53.7/61.5 | **97.2/98.1/92.8** | 95.9/97.3/91.6 |
| switch | 84.3/87.2/77.9 | 85.5/88.6/78.4 | 62.2/66.8/68.6 | 90.1/92.8/83.1 | 73.4/49.4/61.2 | 87.5/93.1/85.0 | **94.8/96.9/91.5** |
| tape | 96.0/95.1/87.6 | 97.2/96.2/89.4 | 49.9/41.1/54.5 | 94.5/93.4/85.9 | 73.9/57.8/66.1 | 97.6/97.3/92.8 | **98.0/98.0/93.7** |
| terminalblock | 89.4/89.7/83.1 | 87.5/89.1/81.0 | 59.8/64.7/68.8 | 83.1/86.2/76.6 | 62.1/36.4/47.8 | 95.0/96.3/90.5 | **98.4/99.0/96.1** |
| toothbrush | 82.0/83.8/77.2 | 78.4/80.1/75.6 | 65.9/70.0/70.1 | 83.7/85.3/79.0 | **91.2**/93.7/**90.9** | 87.0/92.6/84.4 | 90.7/**94.8**/86.9 |
| toy | 69.4/74.2/75.9 | 68.4/75.1/74.8 | 57.8/64.4/73.4 | 70.3/74.8/75.4 | 66.2/57.3/59.8 | 74.6/82.1/83.1 | **89.5/93.2/87.6** |
| toybrick | 63.6/56.1/59.0 | 77.0/71.1/66.2 | 58.3/49.7/58.2 | 73.2/68.7/63.3 | 68.4/45.3/55.9 | 82.5/83.7/72.6 | **85.0/85.5/74.8** |
| transistor1 | 91.0/94.0/85.1 | 93.7/95.9/88.9 | 62.2/69.2/72.1 | 90.2/92.1/84.6 | 73.1/63.1/62.7 | 93.8/97.5/91.8 | **96.2/98.3/92.9** |
| ublock | 89.5/**85.0**/74.2 | 88.8/84.2/**75.5** | 62.4/48.4/51.8 | 80.1/73.9/64.3 | 75.2/68.4/67.9 | 88.5/81.1/72.9 | **90.1**/83.4/74.5 |
| usb | 84.9/84.3/75.1 | 78.7/79.4/69.1 | 57.0/55.3/62.9 | 87.8/88.0/78.3 | 58.9/37.4/45.7 | 92.0/91.2/85.3 | **95.5/94.1/90.2** |
| usbadaptor | 71.1/61.4/62.2 | 76.8/71.3/64.9 | 47.5/38.4/56.5 | 80.1/74.9/67.4 | 76.9/60.2/67.2 | 77.9/75.0/69.3 | **82.6/82.4/72.6** |
| vcpill | 85.1/80.3/72.4 | 87.1/84.0/74.7 | 59.0/48.7/56.4 | 83.8/81.5/69.9 | 64.1/40.4/56.2 | 89.8/88.8/81.6 | **91.4/90.7/82.8** |
| woodenbeads | 81.2/78.9/70.9 | 78.4/77.2/67.8 | 55.1/52.0/60.2 | **82.4**/78.5/73.0 | 62.1/56.4/65.9 | 79.7/**84.9/76.6** | 77.0/83.5/74.9 |
| woodstick | 76.9/61.2/58.1 | 80.8/**72.6/63.6** | 58.2/35.6/45.2 | 80.4/69.2/60.3 | 74.1/66.0/62.1 | 88.9/65.4/63.2 | **92.3**/65.9/60.5 |
| zipper | 95.3/97.2/91.2 | 98.2/98.9/95.3 | 77.2/86.7/77.6 | 96.9/98.1/93.5 | 86.0/87.0/84.0 | 98.8/99.7/97.3 | **99.8/99.9/99.0** |
| Mean | 82.4/79.0/73.9 | 83.0/80.9/74.3 | 57.2/53.4/61.5 | 82.3/79.2/73.2 | 75.6/66.4/69.9 | 86.6/84.9/79.4 | **90.1/88.6/82.8** |

*Table 11.* Comparison with SoTA methods on Real-IAD dataset for multi-class anomaly localization with AU-ROC/AP/F1_max/AU-PRO metrics.

| Normal Indices | RD4AD | UniAD | SimpleNet | DeSTSeg | DiAD | HVQ-Trans | Ours |
|---|---|---|---|---|---|---|---|
| audiojack | 96.6/12.8/22.1/79.6 | 97.6/20.0/31.0/83.7 | 74.4/00.9/04.8/38.0 | 95.5/25.4/31.9/52.6 | 91.6/01.0/03.9/63.3 | 98.5/31.9/41.0/88.0 | 99.0/47.1/51.4/91.5 |
| bottlecap | 99.5/18.9/29.9/95.7 | 99.5/19.4/29.6/96.0 | 85.3/02.3/05.7/45.1 | 94.5/25.3/31.1/25.3 | 94.6/04.9/11.4/73.0 | 98.4/15.6/22.8/90.0 | 99.4/23.4/29.0/95.2 |
| buttonbattery | 97.6/33.8/37.8/86.5 | 96.7/28.5/34.4/77.5 | 75.9/03.2/06.6/40.5 | 98.3/63.9/60.4/36.9 | 84.1/01.4/05.3/66.9 | 99.0/58.1/59.2/85.3 | 99.2/61.0/60.4/91.9 |
| endcap | 96.7/12.5/22.5/89.2 | 95.8/08.8/17.4/85.4 | 63.1/00.5/20.8/25.7 | 89.6/14.4/22.7/29.5 | 81.3/02.0/06.9/38.2 | 95.6/06.0/14.5/84.7 | 97.2/09.2/14.4/91.3 |
| eraser | 99.5/30.8/36.7/96.0 | 99.3/24.4/30.9/94.1 | 80.6/02.7/07.1/42.8 | 95.8/52.7/53.9/46.7 | 91.1/07.7/15.4/67.5 | 99.2/31.6/38.4/92.4 | 99.3/39.5/44.2/93.3 |
| firehood | 98.9/27.7/35.2/87.9 | 98.6/23.4/32.2/85.3 | 70.5/00.3/02.2/25.3 | 97.3/27.1/35.3/34.7 | 91.8/03.2/09.2/66.7 | 98.9/35.0/42.7/93.3 | 98.9/43.3/48.6/94.6 |
| mint | 95.0/11.7/23.0/72.3 | 94.4/07.7/18.1/62.3 | 79.9/00.9/03.6/43.3 | 84.1/10.3/22.4/09.9 | 91.1/05.7/11.6/64.2 | 94.8/17.4/26.8/58.1 | 96.6/29.6/38.7/67.4 |
| mounts | 99.3/30.6/37.1/94.9 | 99.4/28.0/32.8/95.2 | 80.5/02.2/06.8/46.1 | 94.2/30.0/41.3/43.3 | 84.3/00.4/01.1/48.8 | 99.6/30.2/37.0/97.7 | 99.7/39.2/40.0/98.6 |
| pcb | 97.5/15.8/24.3/88.3 | 97.0/18.5/28.1/81.6 | 78.0/01.4/04.3/41.3 | 97.2/37.1/40.4/48.8 | 92.0/03.7/07.4/66.5 | 97.7/28.7/37.2/84.1 | 99.0/48.8/51.2/92.1 |
| phonebattery | 77.3/22.6/31.7/94.5 | 85.5/11.2/21.6/88.5 | 43.4/00.1/00.9/11.8 | 79.5/25.6/33.8/39.5 | 96.8/05.3/11.4/85.4 | 98.0/24.1/31.5/87.3 | 99.2/41.1/44.5/94.0 |
| plasticnut | 98.8/21.1/29.6/91.0 | 98.4/20.6/27.1/88.9 | 77.4/00.6/03.6/41.5 | 96.5/44.8/45.7/38.4 | 81.1/00.4/03.4/38.6 | 97.0/16.2/26.8/84.9 | 98.3/27.2/31.1/90.1 |
| plasticplug | 99.1/20.5/28.4/94.9 | 98.6/17.4/26.1/90.3 | 78.6/00.7/01.9/38.8 | 91.9/20.1/27.3/21.0 | 92.9/08.7/15.0/66.1 | 99.2/23.6/29.7/95.0 | 99.5/37.1/41.3/97.1 |
| porcelaindoll | 99.2/24.8/34.6/95.7 | 98.7/14.1/24.5/93.2 | 81.8/02.0/06.4/47.0 | 93.1/35.9/40.3/24.8 | 93.1/01.4/04.8/70.4 | 97.9/11.4/18.6/89.9 | 98.8/18.3/26.3/93.8 |
| regulator | 98.0/07.8/16.1/88.6 | 95.5/09.1/17.4/76.1 | 76.6/00.1/00.6/38.1 | 88.8/18.9/23.6/17.5 | 84.2/00.4/01.5/44.4 | 98.0/07.0/16.2/89.7 | 99.7/37.4/42.2/98.6 |
| rolledstripbase | 99.7/31.4/39.9/98.4 | 99.6/20.7/32.2/97.8 | 80.5/01.7/05.1/52.1 | 99.2/48.7/50.1/55.5 | 87.7/00.6/03.2/63.4 | 98.9/16.1/25.9/96.2 | 99.7/32.4/42.5/98.9 |
| simcardset | 98.5/40.2/44.2/89.5 | 97.9/31.6/39.8/85.0 | 71.0/06.8/14.3/30.8 | 99.1/65.5/62.1/73.9 | 89.9/01.7/05.8/60.4 | 99.1/39.7/43.2/93.9 | 99.3/48.9/50.1/95.4 |
| switch | 94.4/18.9/26.6/90.9 | 98.1/33.8/40.6/90.7 | 71.7/03.7/09.3/44.2 | 97.4/57.6/55.6/44.7 | 90.5/01.4/05.3/64.2 | 99.0/51.5/55.2/91.5 | 99.5/63.6/63.4/95.8 |
| tape | 99.7/42.4/47.8/98.4 | 99.7/29.2/36.9/97.5 | 77.5/01.2/03.9/41.4 | 99.0/61.7/57.6/48.2 | 81.7/00.4/02.7/47.3 | 99.6/20.5/29.8/98.3 | 99.7/29.8/36.4/98.8 |
| terminalblock | 99.5/27.4/35.8/97.6 | 99.2/23.1/30.5/94.4 | 87.0/00.8/03.6/54.8 | 96.6/40.6/44.1/34.8 | 75.5/00.1/01.1/38.5 | 99.6/35.5/39.3/97.1 | 99.8/48.3/51.0/98.9 |
| toothbrush | 96.9/26.1/34.2/88.7 | 95.7/16.4/25.3/84.3 | 84.7/07.2/14.8/52.2 | 94.3/30.0/37.3/42.8 | 82.0/01.9/06.6/54.5 | 98.4/37.2/44.4/90.6 | 98.8/39.8/47.9/93.8 |
| toy | 95.2/05.1/12.8/82.3 | 93.4/04.6/12.4/70.5 | 67.7/00.1/00.4/25.0 | 86.3/08.1/15.9/16.4 | 82.1/01.1/04.2/50.3 | 94.2/05.4/10.4/82.2 | 97.8/19.8/25.5/90.8 |
| toybrick | 96.4/16.0/24.6/75.3 | 97.4/17.1/27.6/81.3 | 86.5/05.2/11.1/56.3 | 94.7/24.6/30.8/45.5 | 93.5/03.1/08.1/66.4 | 97.5/28.9/37.3/82.7 | 98.6/44.3/48.7/89.6 |
| transistor1 | 99.1/29.6/35.5/95.1 | 98.9/25.6/33.2/94.3 | 71.7/05.1/11.3/35.3 | 97.3/43.8/44.5/45.4 | 88.6/07.2/15.3/58.1 | 98.1/27.1/31.8/91.4 | 99.1/40.2/43.6/95.9 |
| ublock | 99.6/40.5/45.2/96.9 | 99.3/22.3/29.6/94.3 | 76.2/04.8/12.2/34.0 | 96.9/57.1/55.7/38.5 | 88.8/01.6/05.4/54.2 | 99.2/19.0/27.1/94.1 | 99.5/24.2/35.6/97.8 |
| usb | 98.1/26.4/35.2/91.0 | 97.9/20.6/31.7/85.3 | 81.1/01.5/04.9/52.4 | 98.4/42.2/47.7/57.1 | 78.0/01.0/03.1/28.0 | 99.2/29.0/38.1/93.6 | 99.6/43.4/48.3/97.0 |
| usbadaptor | 94.5/09.8/17.9/73.1 | 96.6/10.5/19.0/78.4 | 67.9/00.2/01.3/28.9 | 94.9/25.5/34.9/36.4 | 94.0/02.3/06.6/75.5 | 94.5/11.8/21.1/73.0 | 96.8/18.1/27.3/84.2 |
| vcpill | 98.3/43.1/48.6/88.7 | 99.1/40.7/43.0/91.3 | 68.2/01.1/03.3/22.0 | 97.1/64.7/62.3/42.3 | 90.2/03.5/25.2/60.8 | 99.1/61.9/63.3/92.0 | 99.0/58.4/61.2/92.6 |
| woodenbeads | 98.0/27.1/34.7/85.7 | 97.6/16.5/23.6/84.6 | 68.1/02.4/06.0/28.3 | 94.7/38.9/42.9/39.4 | 85.0/01.1/04.7/45.6 | 96.6/21.5/30.0/77.2 | 97.3/26.2/31.4/83.1 |
| woodstick | 97.8/30.7/38.4/85.0 | 94.0/36.2/44.3/77.2 | 76.1/01.4/06.0/32.0 | 97.9/60.3/60.0/51.0 | 90.9/02.6/08.0/60.7 | 97.8/47.3/50.3/91.4 | 98.4/48.5/51.9/93.1 |
| zipper | 99.1/44.7/50.2/96.3 | 98.4/32.5/36.1/95.1 | 89.9/23.3/31.2/55.5 | 98.2/35.3/39.0/78.5 | 90.2/12.5/18.8/53.5 | 98.5/37.3/43.6/94.6 | 99.0/43.8/49.7/97.4 |
| Mean | 97.3/25.0/32.7/89.6 | 97.3/21.1/29.2/86.7 | 75.7/02.8/06.5/39.0 | 94.6/37.9/41.7/40.6 | 88.0/02.9/07.1/58.1 | 98.0/27.6/34.4/88.7 | 98.9/37.7/42.6/93.1 |

## H. Quantitative Analysis of False Positive Rates on Benchmark Datasets:

To further demonstrate the effectiveness of OmiAD, we calculated the False Positive Rate (FPR) for OmiAD, HVQ-Trans, UniAD, and DiAD across four benchmark datasets: MVTec-AD, VisA, MPDD, and Real-IAD, as shown in Table 12. In our experiments, the threshold for calculating the FPR was set based on the number of anomaly samples. Specifically, we ranked the samples by their anomaly scores and set the threshold to the score at the position corresponding to the number of anomaly samples. This method ensures that false positives are calculated consistently across the different models.

The results consistently show that OmiAD achieves a significantly lower FPR than its counterparts, highlighting its superior anomaly detection capabilities.

For instance, OmiAD achieves an FPR of 2.50% on MVTec-AD, outperforming HVQ-Trans (3.54%), UniAD (6.12%), and DiAD (8.70%). Similar improvements are observed on other datasets, with OmiAD achieving 9.50% on VisA, 10.71% on MPDD, and 18.22% on Real-IAD. These substantial reductions in FPR demonstrate OmiAD's ability to minimize false alarms effectively while maintaining high anomaly detection accuracy.

This performance advantage stems from OmiAD's global modeling strategy, which leverages the adaptive masking (AM) technique to mitigate shortcut learning by reducing reliance on spurious correlations or local patterns—common sources of false positives. The adaptive masking dynamically guides the model to focus on global contextual information, improving anomaly reconstruction and reducing false alarms. Such a capability is critical in real-world industrial applications, where a low FPR ensures model reliability and minimizes unnecessary interventions caused by false alarms.

## I. More Ablation Studies:

### I.1. Ablation Study on Initial Time Step $t_{\text{init}}$:

We conduct an ablation study on the MVTec-AD dataset to assess the effect of the initial diffusion step $t_{\text{init}}$ on the performance of OmiAD. As shown in Table 13, the model maintains stable and high performance across a broad range of

*Table 12.* False Positive Rate (FPR) on four datasets. The values are in percentages (%).

| Dataset | HVQ-Trans | UniAD | DiAD | Ours |
|---------|-----------|-------|------|------|
| **MVTec-AD** | 3.54 | 6.12 | 8.70 | **2.50** |
| **VisA** | 14.58 | 14.50 | 23.67 | **9.50** |
| **MPDD** | 19.22 | 19.52 | 23.07 | **10.71** |
| **Real-IAD** | 22.67 | 24.00 | 34.94 | **18.22** |

$t_{\text{init}}$ values (from 800 to 960). However, when $t_{\text{init}}$ is set to 1000, a noticeable performance degradation occurs. This drop is attributed to the excessive noise introduced at this step, which hinders effective reconstruction and impairs accurate anomaly localization. Based on our findings, we recommend setting $t_{\text{init}} = 960$, as it achieves the best performance at both the image and pixel levels, striking a balance between reconstruction quality and anomaly detection accuracy.

*Table 13.* Ablation study results on different initial diffusion steps $t_{\text{init}}$.

| $t_{\text{init}}$ | 600 | 700 | 800 | 940 | 960 | 980 | 1000 |
|-------------------|-----|-----|-----|-----|-----|-----|------|
| Image AUROC | 97.7 | 98.1 | 98.3 | 98.7 | **98.8** | 95.9 | 73.5 |
| Pixel AUROC | 96.9 | 97.1 | 97.4 | 97.7 | **97.7** | 97.3 | 80.0 |

### I.2. Ablation Study on Architecture Robustness in Diffusion Models:

To assess the architectural flexibility of our approach, we conduct experiments on the MVTec-AD dataset, evaluating OmiAD with different architectures for the diffusion model. Specifically, we replace the default U-Net (Ronneberger et al., 2015) with DiT (Peebles & Xie, 2023) and U-ViT (Bao et al., 2023) as the backbone for the diffusion model. As shown in Table 14, OmiAD consistently demonstrates strong and stable performance across all three architectures, with minimal variation in both image-level and pixel-level AUROC metrics. These results underscore the robustness of our method, highlighting its ability to generalize effectively across different architectures used in the diffusion model.

*Table 14.* Ablation study on the diffusion model architecture.

| Model | DiT | U-ViT | U-Net |
|-------|-----|-------|-------|
| Image AUROC | 98.2 | 98.4 | **98.8** |
| Pixel AUROC | 97.5 | 97.4 | **97.7** |

### I.3. Ablation Study on Backbone Feature Extractor:

We conduct ablation experiments on the MVTec-AD dataset to evaluate the impact of different feature extractors on OmiAD's performance. Specifically, we replace the default EfficientNet (Tan & Le, 2019) backbone with ResNet (He et al., 2016) variants of increasing depth: ResNet-18, ResNet-34, ResNet-50, and ResNet-101. For the ResNet-based models, we set the initial diffusion step $t_{\text{init}}$ to 400, in contrast to $t_{\text{init}}$ to 960 for the EfficientNet backbone. As reported in Table 15, all models demonstrate strong performance, highlighting the robustness of our framework across different feature extractors. These results confirm that while OmiAD is flexible in backbone selection, EfficientNet achieves the best balance between semantic richness and localization accuracy.

## J. More Qualitative Results:

To further enhance the comparative analysis between OmiAD and existing methodologies, we present a comprehensive visual exploration of anomaly detection results involving OmiAD, HVQ-Trans, DiAD, and UniAD, as shown in Fig. 4 for MVTec-AD, Fig. 5 for VisA, Fig. 6 for MPDD, Fig. 7 and Fig. 8 for Real-IAD. Benefiting from the global modeling capability of OmiAD's masking strategy, the model effectively reduces shortcut learning, thereby achieving more accurate anomaly localization and significantly reducing false positives.

*Table 15.* Ablation study on the impact of different feature extractors.

| Backbone | ResNet-18 | ResNet-34 | ResNet-50 | ResNet-101 | EfficientNet |
|---|---|---|---|---|---|
| Image AUROC | 97.3 | 97.2 | 95.3 | 94.2 | **98.8** |
| Pixel AUROC | 97.3 | 97.5 | 97.3 | 97.3 | **97.7** |

Additionally, we provide detailed visualizations of OmiAD's anomaly detection results across these datasets. The results, showcased in Fig. 9 for MVTec-AD, Fig. 10 for VisA, Fig. 11 for MPDD, and Fig . 12 and Fig. 13 for Real-IAD, highlight the outstanding performance of OmiAD. In these visualizations, OmiAD consistently demonstrates the ability to successfully transform anomalous samples into their corresponding normal counterparts. These results not only reflect the model's accuracy in identifying and localizing anomalous regions but also showcase its robustness in handling diverse types of anomalies, including object anomalies and texture damages.

By precisely capturing reconstruction differences, OmiAD effectively identifies and highlights regions that deviate from the normal distribution. These visual insights underscore the model's capability to accurately discern abnormal regions, further validating OmiAD as a powerful and versatile solution for anomaly detection and localization across a wide range of datasets.

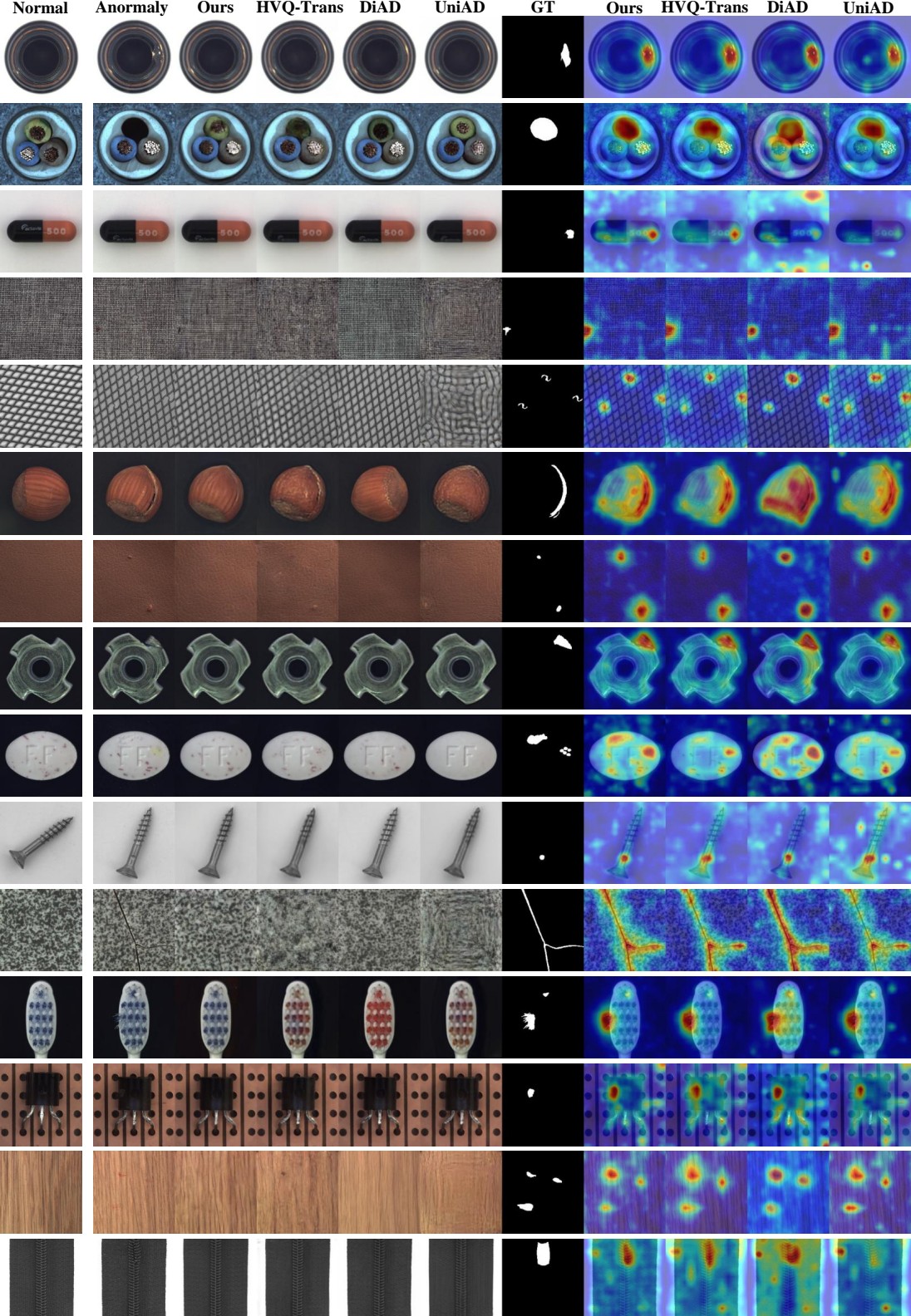

*Figure 4.* Qualitative results for anomaly localization on MVTec-AD. From left to right: normal sample as the reference, anomaly, our reconstruction, HVQ-Trans reconstruction, DiAD reconstruction, UniAD reconstruction, ground-truth, our predicted anomaly map, HVQ-Trans predicted anomaly map, DiAD predicted anomaly map and UniAD predicted anomaly map.

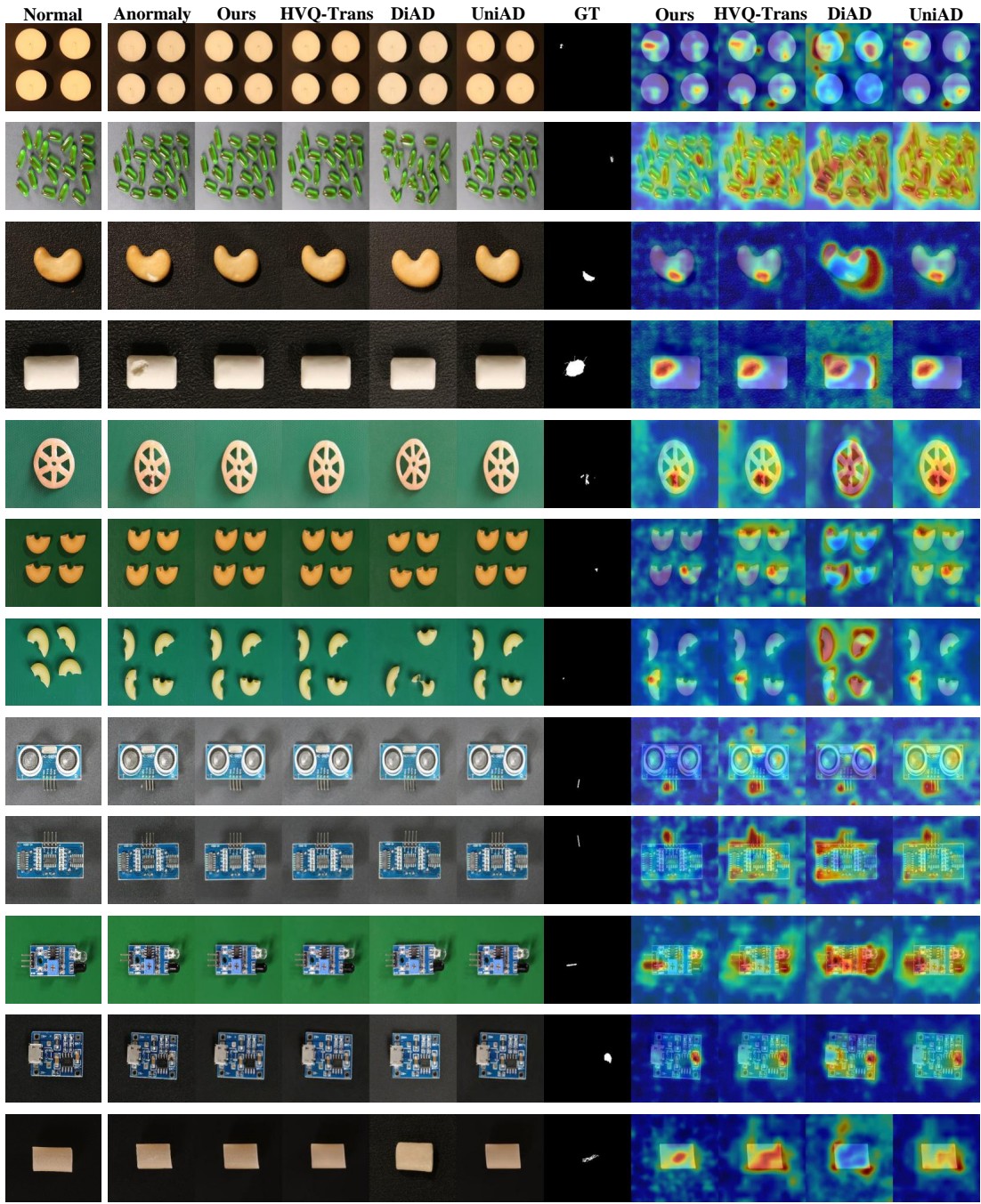

*Figure 5.* Qualitative results for anomaly localization on VisA. From left to right: normal sample as the reference, anomaly, our reconstruction, HVQ-Trans reconstruction, DiAD reconstruction, UniAD reconstruction, ground-truth, our predicted anomaly map, HVQ-Trans predicted anomaly map, DiAD predicted anomaly map and UniAD predicted anomaly map.

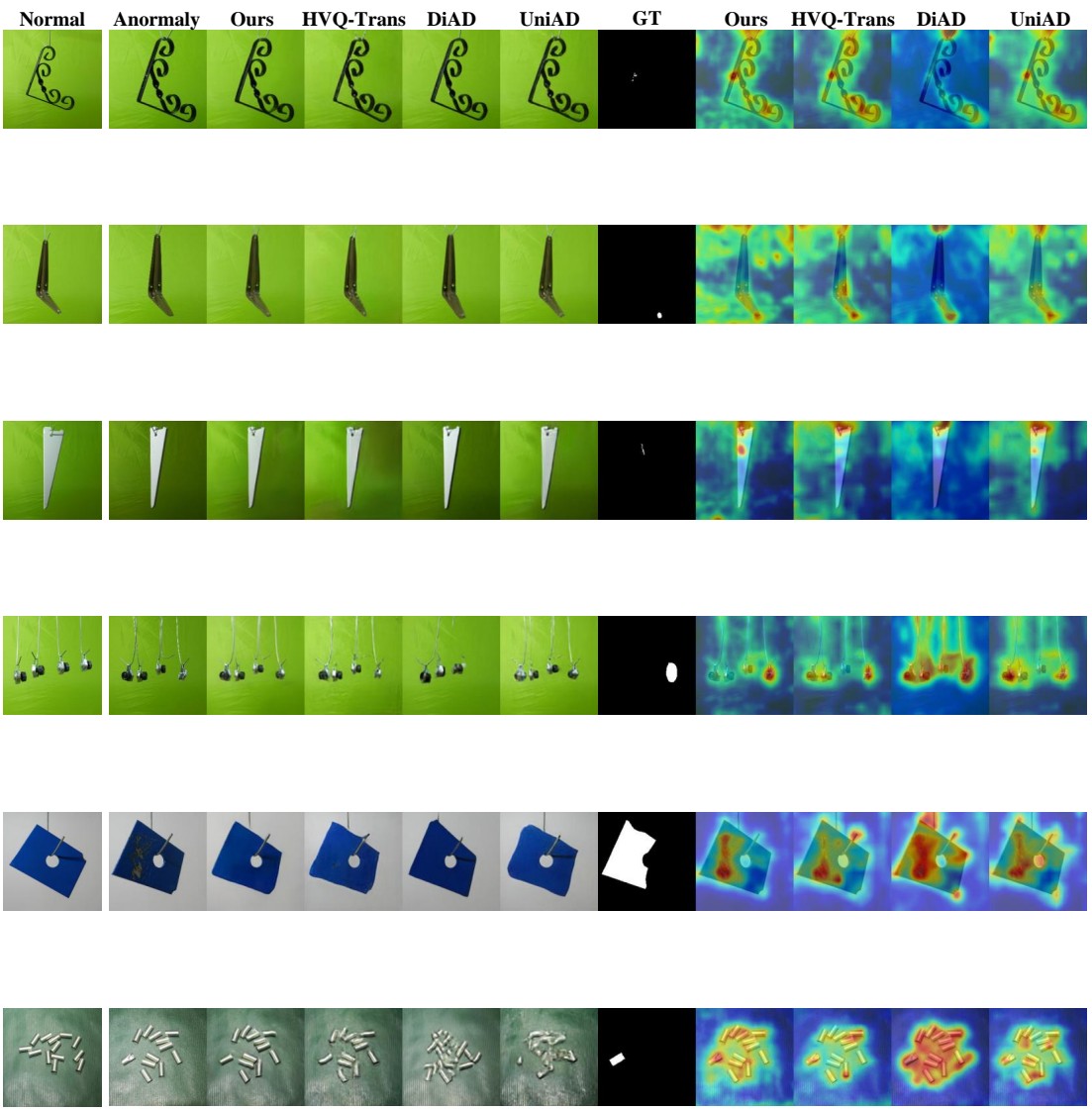

*Figure 6.* Qualitative results for anomaly localization on MPDD. From left to right: normal sample as the reference, anomaly, our reconstruction, HVQ-Trans reconstruction, DiAD reconstruction, UniAD reconstruction, ground-truth, our predicted anomaly map, HVQ-Trans predicted anomaly map, DiAD predicted anomaly map and UniAD predicted anomaly map.

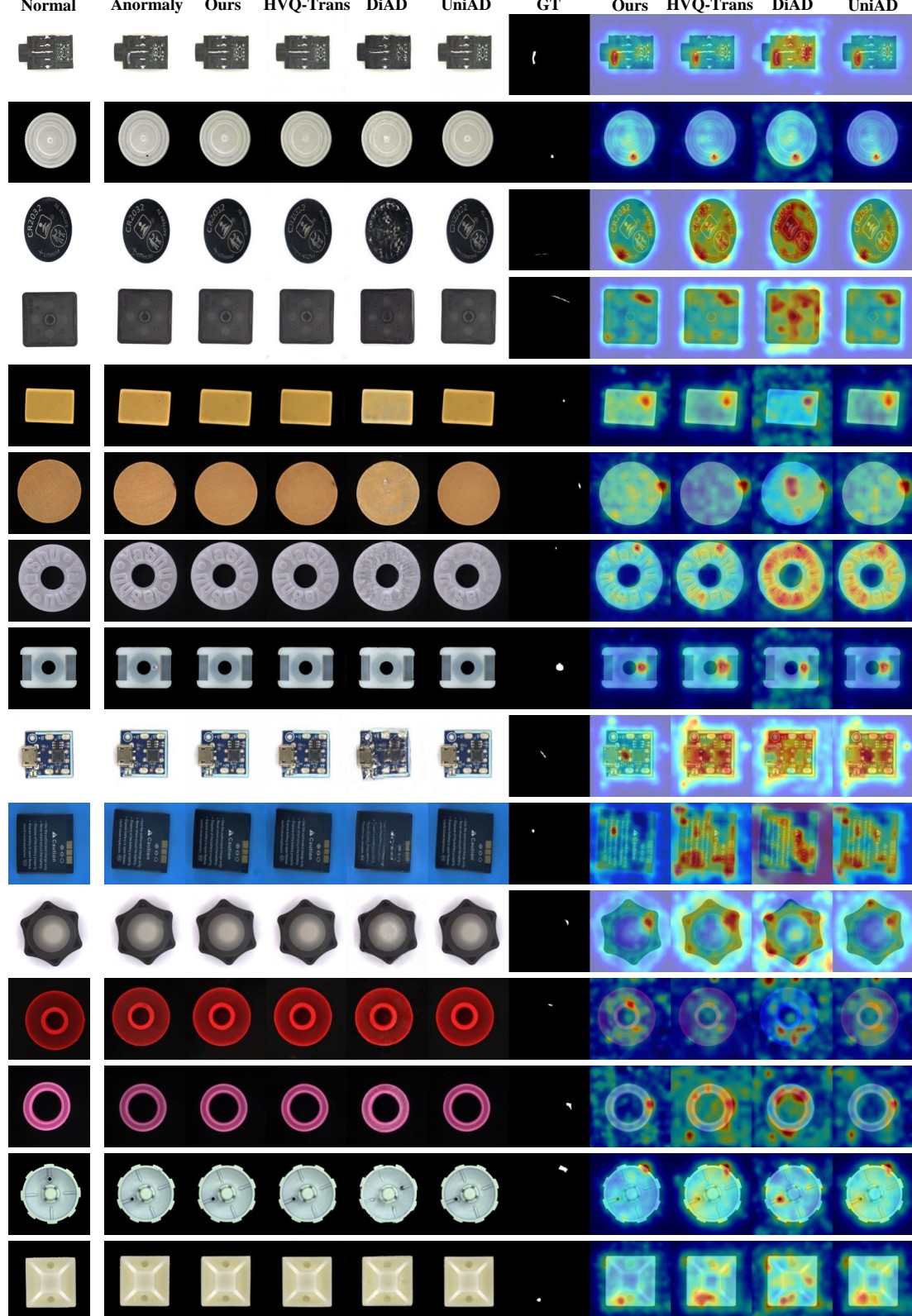

Figure 7. Qualitative results for anomaly localization on Real-IAD(Part 1). From left to right: normal sample as the reference, anomaly, our reconstruction, HVQ-Trans reconstruction, DiAD reconstruction, UniAD reconstruction, ground-truth, our predicted anomaly map, HVQ-Trans predicted anomaly map, DiAD predicted anomaly map and UniAD predicted anomaly map.

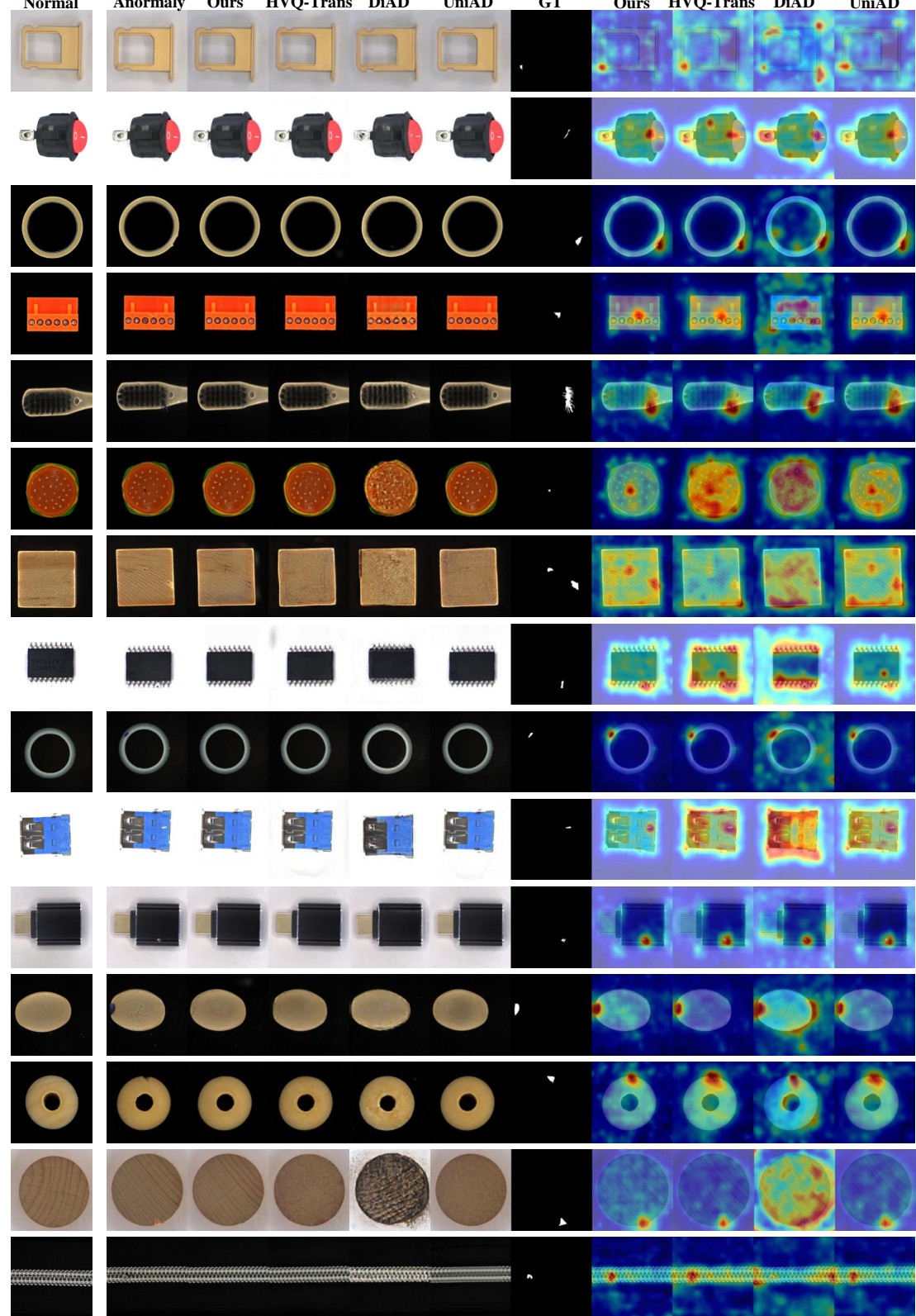

*Figure 8.* Qualitative results for anomaly localization on Real-IAD(Part 2). From left to right: normal sample as the reference, anomaly, our reconstruction, HVQ-Trans reconstruction, DiAD reconstruction, UniAD reconstruction, ground-truth, our predicted anomaly map, HVQ-Trans predicted anomaly map, DiAD predicted anomaly map and UniAD predicted anomaly map.

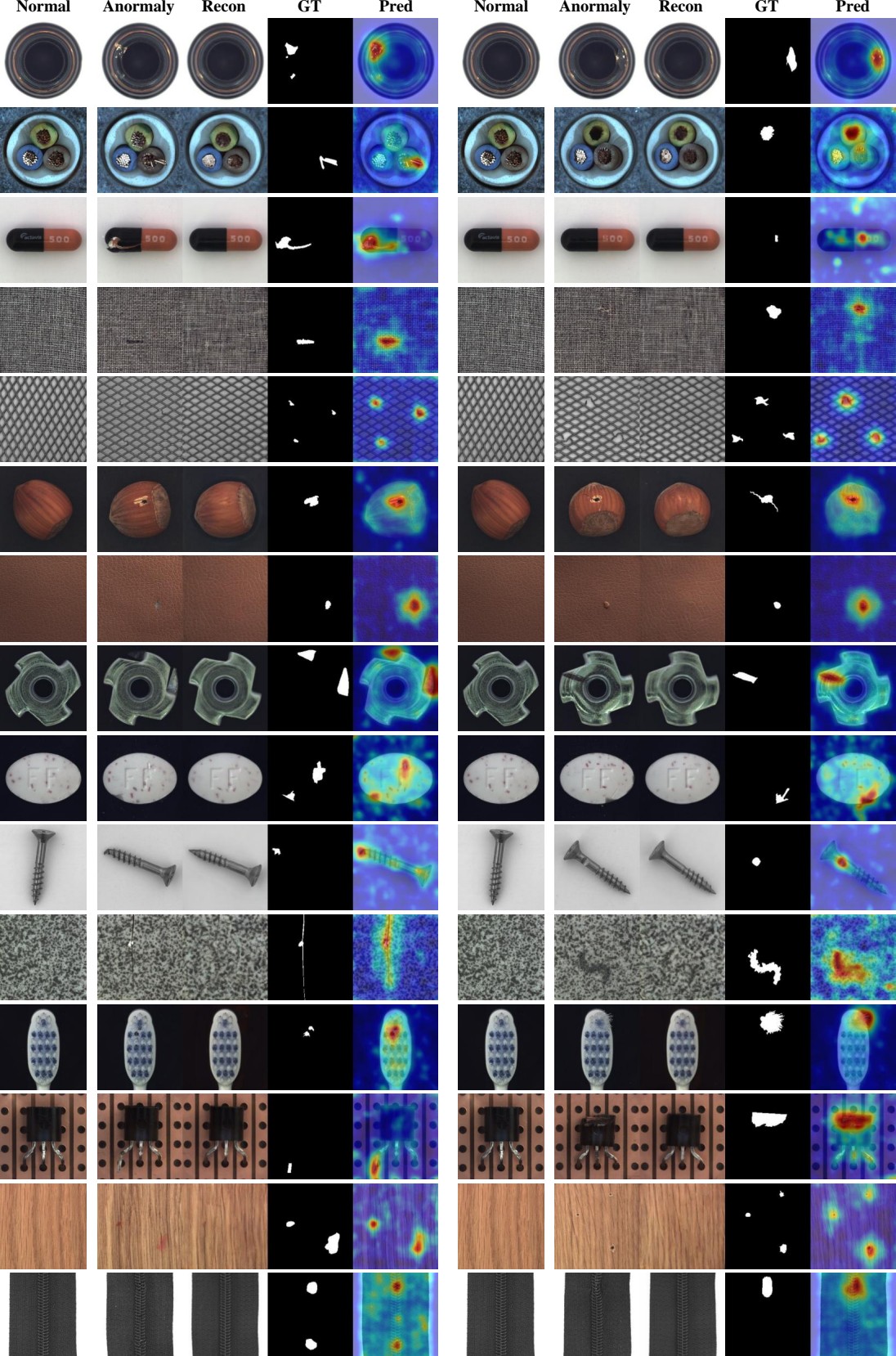

*Figure 9.* Qualitative results for anomaly localization on MVTec-AD. From left to right: normal sample as the reference, anomaly, our reconstruction, ground-truth, and our predicted anomaly map.

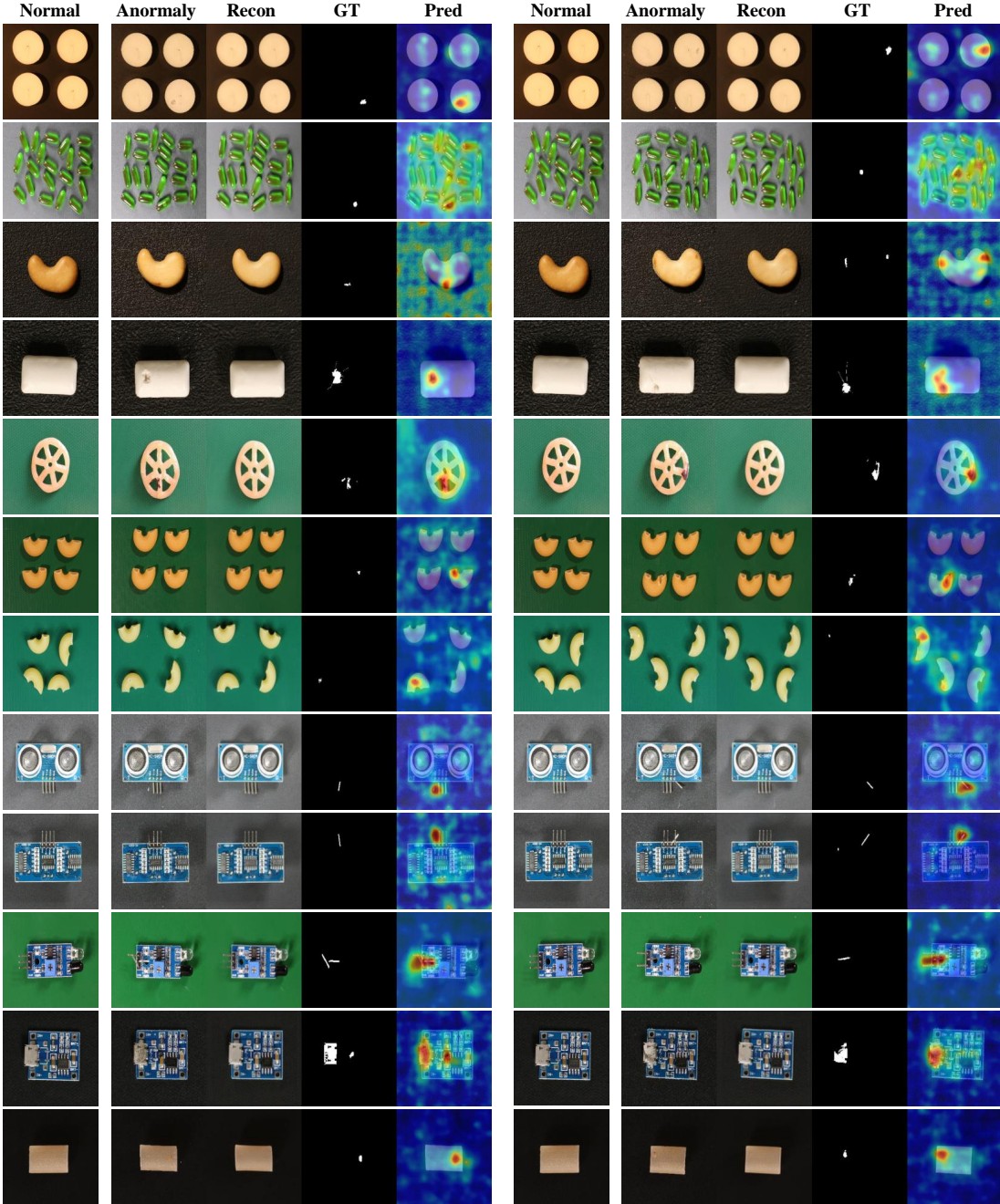

*Figure 10.* Qualitative results for anomaly localization on VisA. From left to right: normal sample as the reference, anomaly, our reconstruction, ground-truth, and our predicted anomaly map.

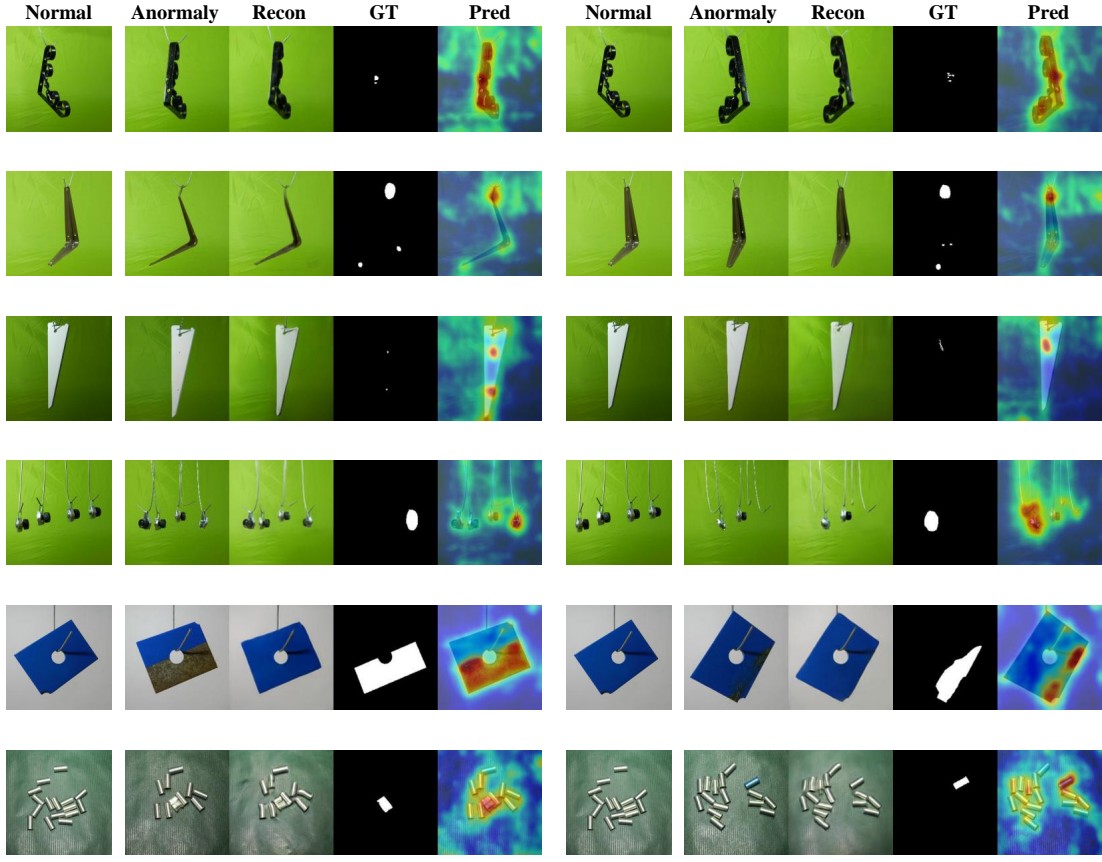

*Figure 11.* Qualitative results for anomaly localization on MPDD. From left to right: normal sample as the reference, anomaly, our reconstruction, ground-truth, and our predicted anomaly map.

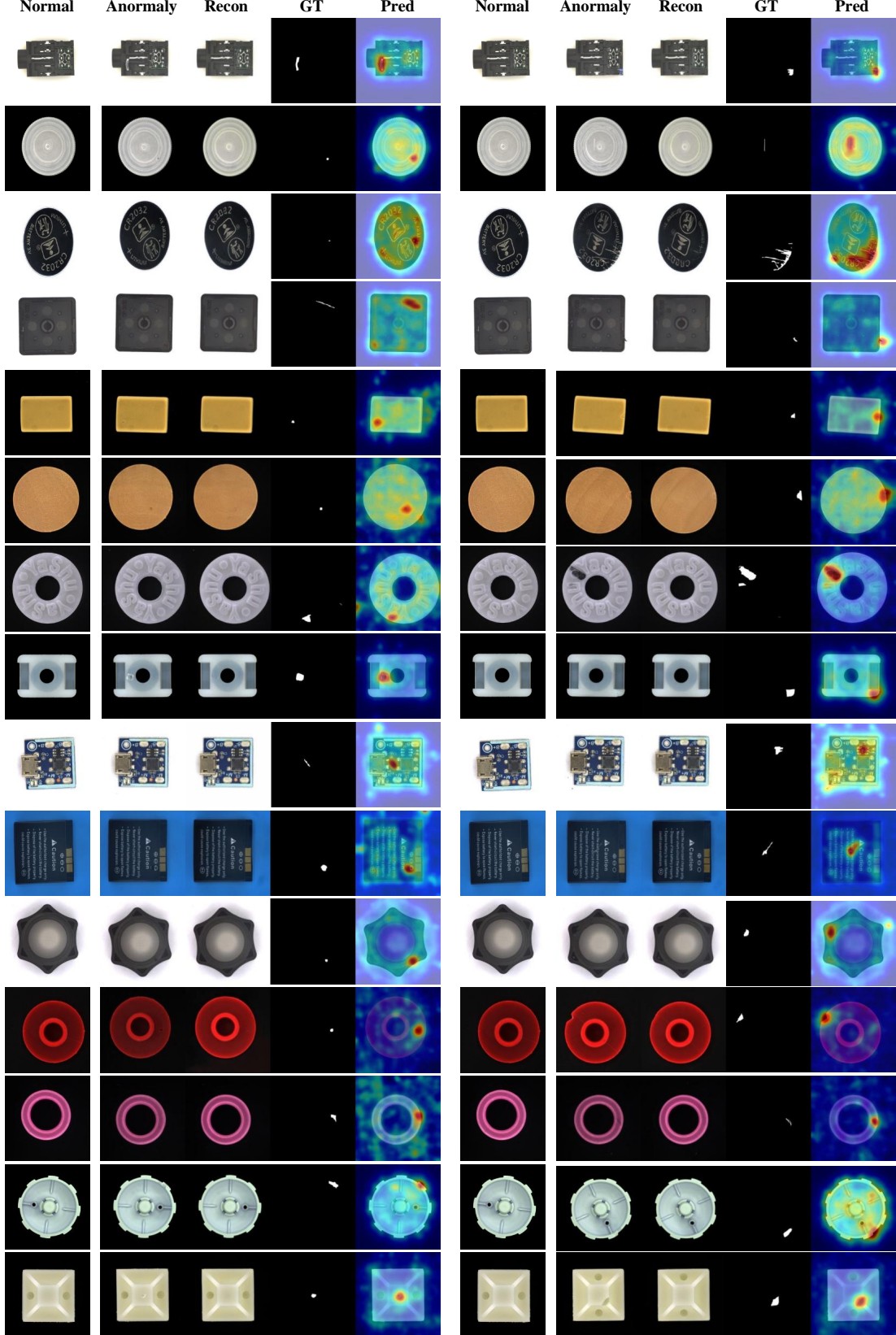

*Figure 12.* Qualitative results for anomaly localization on Real-IAD(Part 1). From left to right: normal sample as the reference, anomaly, our reconstruction, ground-truth, and our predicted anomaly map.

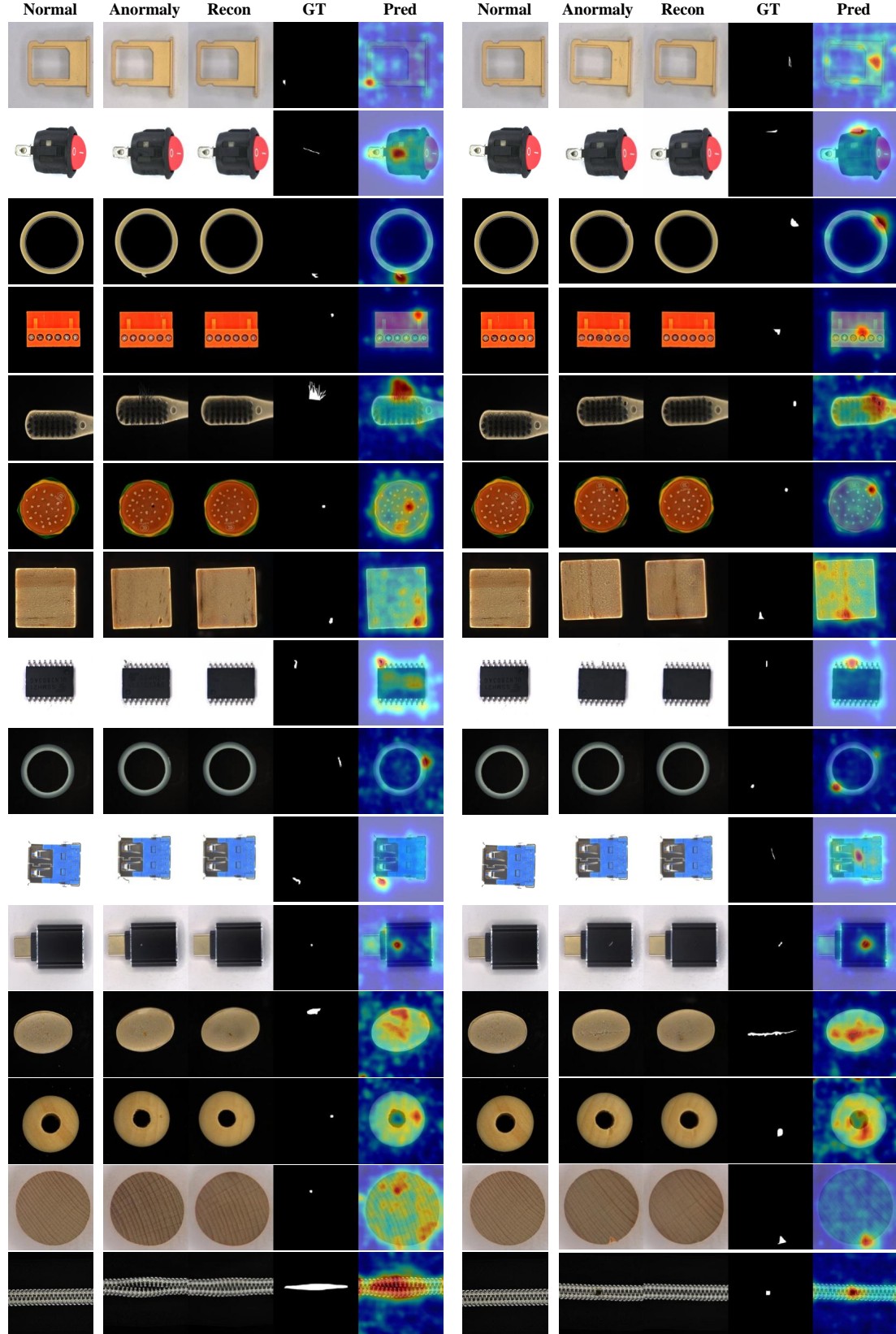

*Figure 13.* Qualitative results for anomaly localization on Real-IAD(Part 2). From left to right: normal sample as the reference, anomaly, our reconstruction, ground-truth, and our predicted anomaly map.

