# OpenReview forum: "OmiAD: One-Step Adaptive Masked Diffusion Model for Multi-class Anomaly Detection via Adversarial Distillation"
_ICML.cc/2025/Conference — ICML 2025 poster_

### Official Review · Reviewer_96ub · 2025-03-11

**Overall Recommendation:** 3

**Summary:**

The paper introduces OmiAD, a one-step adaptive masked diffusion model for multi-class anomaly detection (MUAD).

The authors propose Adaptive Masking Diffusion Model (AMDM) to mitigate "identical shortcut" issues by dynamically adjusting mask ratios based on noise levels and Adversarial Score Distillation (ASD) to compress multi-step diffusion processes into a single inference step.

## Update after rebuttal

The authors have addressed most of my concerns. I remain "weak accept".

It is worth pointing out that "Inference Time" is decided by the algorithm and GPU, which is irrelevant to dataset. I understant that you run the speed test on several dataset. But you only need to report their average. Report Inference Time for each dataset is wierd.

**Claims And Evidence:**

CV methodology paper. No claims apart from the claim of superiority.

**Essential References Not Discussed:**

Some recent MUAD methods are not compared, including MambaAD (NIPS24), ViTAD(Arxiv24), ReContrast (NIPS23), etc.

**Experimental Designs Or Analyses:**

The experiments are sound. The setting follows common MUAD setting.

**Methods And Evaluation Criteria:**

The proposed method make sense.

**Other Comments Or Suggestions:**

NO

**Other Strengths And Weaknesses:**

Strengths:

1. OmiAD reduces inference time drastically compared to other Diffusion-based UAD methods, making it viable for real-time industrial applications.

2. Extensive experiments, works well across different datasets.

Weaknesses:

1. Several recent Multi-class Unsupervised Anomaly Detection (MUAD) methods appear to be missing from the comparison, including MambaAD (NeurIPS 2024), ViTAD (arXiv 2024), and ReContrast (NeurIPS 2023), etc. Their performances are relatively comparable to the results of this work.

2. In Table 2, why SimpleNet spend 18 seconds for a bach? It is extremely unreasonable. SimpleNet only consists of a CNN backbone and a lightweight head. It should be faster than RD4AD. Furthermore, why do different datasets have different inference speed? They should be the same. It is also not clear whether the time is for one image or one batch.

3. A figure that depicts the overall method is favored. Figure 1 only presents ASD, which is only a part of the proposed method.

4. The results of single-class UAD should be presented as a reference.

5. This article is application-oriented and has a relatively narrow domain. I am not sure if it aligns with ICML's interests.

**Questions For Authors:**

NO

**Relation To Broader Scientific Literature:**

Related to Diffusion-based UAD methods, such as DiAD.

**Theoretical Claims:**

None.

---

> ### Author Rebuttal · Authors · 2025-03-31
>
> We are grateful for the time you spent reviewing our paper in detail. Your insightful comments have been extremely helpful, and we deeply appreciate your input.
>
> **Q1: Comparison with Recent Multi-class UAD Methods**
>
> A1：We have compared our results on the MVTEC and VISA datasets with MambaAD, ViTAD, and ReContrast, as summarized in the table below. All results are sourced from the original publications. As shown, OmiAD consistently achieves superior performance across datasets.
> | Method|**MVTec**||**VisA**||
> |-|-|-|-|-|
> ||Image AUROC|Pixel AUROC|Image AUROC|Pixel AUROC|
> |ReContrast|98.2|–|95.1|–|
> |MambaAD|98.6|**97.7**|94.3|98.5|
> |ViTAD|98.3|**97.7**|90.5|98.2|
> |**OURS**|**98.8**|**97.7**|**95.3**|**98.9**|
>
> **Q2: Speed of SimpleNet**
>
> A2: We revisited the official SimpleNet implementation and found that it upsamples both the anomaly score map and feature map to a shape of batch × 1536 × 288 × 288. The resulting tensor is then transferred from GPU to CPU and stored as a NumPy array (as seen in line 113 of the source code in common.py). This operation accounts for the majority of the inference time, resulting in significantly slower inference. Since our evaluation was based on the official public code, we observed these slower inference times accordingly.
>
> Based on your valuable suggestion, we optimized the implementation by removing the unnecessary upsampling step and the conversion to NumPy arrays, and streamlined the overall code. This optimization resulted in inference speeds that are consistent with the expected performance of the embedding-based structure and are lower than most methods reported in Table 2 of our paper. The updated inference times on the four datasets are shown below. As observed, the inference time remains longer than that of OmiAD, and thus does not affect the conclusions presented in our paper.
> |Dataset|MVTEC|VISA|MPDD|REALIAD|
> |-|-|-|-|-|
> |Inference Time (s)|0.0292|0.0296|0.0281|0.0277|
>
> **Q3: Overall method**
>
> A3: Should the paper be accepted, we will include a comprehensive illustration of the overall method in the camera-ready version, covering all components of OmiAD, including both AMDM and ASD modules.
>
> **Q4：Single-class UAD results**
>
> A4: We present the single-class anomaly detection performance on the MVTec and MPDD datasets as follows. These results will be detailed in the appendix for further reference. The single-class results on VisA and Real-IAD are currently under evaluation and will be included in the camera-ready version.
>
> **Single class UAD results for MVTec dataset：**
> |Metric|bottle|cable|capsule|hazelnut|metal_nut|pill|screw|toothbrush|transistor|zipper|carpet|grid|leather|tile|wood|**mean**|
> |-|-|-|-|-|-|-|-|-|-|-|-|-|-|-|-|-|
> |Image AUROC|100.0|99.2|96.8|100.0|99.4|97.0|94.7|99.4|99.9|99.7|100.0|100.0|100.0|100.0|98.6|**99.0**|
> |Pixel AUROC|98.6|98.3|98.9|98.8|97.0|97.2|99.4|98.8|98.9|98.2|98.8|98.6|99.3|92.7|94.2|**97.8**|
>
> **Single-class UAD results for MPDD dataset：**
> |Metric|bracket_black|bracket_brown|bracket_white|connector|metal_plate|tubes|**mean**|
> |-|-|-|-|-|-|-|-|
> |Image AUROC|96.3|99.1|93.2|96.7|100.0|94.8|**96.7**|
> |Pixel AUROC|98.8|98.3|98.6|98.9|98.5|98.8|**98.7**|
>
> **Q5: Application-oriented focus and alignment with ICML's interests**
>
> A5: Thank you for your comment. While this work is focused on multi-class anomaly detection, we did not make innovations in areas like anomaly score calculation or anomaly classification, which are task-specific. Instead, our innovation lies in enhancing generative capabilities and inference speed, which are more general improvements that are highly relevant in the current machine learning field. In the context of anomaly detection, the proposed method improves detection performance by enhancing the completion of anomalies with the powerful generator.
>
> Besides, the core contribution of our work is a novel one-step adaptive masked diffusion model, which integrates a random masking strategy into the one-step diffusion process to enhance generation quality and efficiency. This design offers broader applicability to tasks demanding efficient and high-quality generation. Furthermore, the rapid and efficient generation characteristics of the proposed method are closely aligned with the challenges currently faced in industrial anomaly detection. Thus, we have applied and validated the capabilities of our method in this task. Our work aligns closely with ICML's focus on advancing machine learning techniques with practical applicability.
>
> Finally, anomaly detection is also an important direction of machine learning, and it has strong correlation with unsupervised machine learning, statistical machine learning and so on. In recent years, many related papers have been published at ICML.
>
> **Due to character limitations, we would be happy to discuss and address any remaining questions during the discussion phase.**

---

### Official Review · Reviewer_3S6t · 2025-03-12

**Overall Recommendation:** 4

**Summary:**

To address the slow inference speed due to the iterative denoising nature of the diffusion model, this paper proposes a one-step masked diffusion model for multi-class anomaly detection, OmiAD, which uses a multi-step Adaptive Masked Diffusion Model (ADM) with compression using ASD. State-of-the-art performance is achieved on all seven metrics for four different datasets, along with a significant improvement in inference speed.

**Claims And Evidence:**

The claims made in the submission are well-supported by clear and convincing evidence. The authors provide extensive experimental results on four diverse datasets, demonstrating the effectiveness of OmiAD in terms of both anomaly detection and localization. The proposed method shows significant improvements over existing approaches, which strongly supports the claims of enhanced performance and efficiency. The ablation studies further validate the contributions of individual components like the adaptive masking strategy and the adversarial score distillation.

**Essential References Not Discussed:**

The paper cites relevant previous work on diffusion models, anomaly detection, and distillation methods.

**Experimental Designs Or Analyses:**

The experimental designs are sound and comprehensive. The authors conducted experiments on four diverse datasets, comparing OmiAD with several baseline methods. The results demonstrate the superiority of OmiAD in terms of both detection performance and inference speed. The ablation studies provide insights into the contributions of different components of the proposed method. The visualizations of reconstruction results and anomaly maps further support the effectiveness of OmiAD in localizing anomalies.

**Methods And Evaluation Criteria:**

The proposed OmiAD method makes sense for the problem of multi-class anomaly detection, where efficiency and accuracy are crucial. The use of a diffusion model with an adaptive masking strategy and adversarial score distillation is appropriate for addressing the challenges of shortcut learning and slow inference. The evaluation criteria, including AU-ROC, AP, F1 max, and AU-PRO, are standard and suitable for assessing the performance of anomaly detection methods.

**Other Comments Or Suggestions:**

The paper is well-written and well-structured, making it easy to follow the methodology and experimental results. However, providing more details on the implementation of the adaptive masking strategy and the adversarial distillation process would be beneficial for readers interested in reproducing the results.

**Other Strengths And Weaknesses:**

Strengths:
The proposed OmiAD method effectively addresses the challenges of shortcut learning and slow inference in diffusion-based anomaly detection.
The extensive experimental results on multiple datasets demonstrate the robustness and versatility of OmiAD.
The adversarial score distillation approach offers a novel way to compress multi-step diffusion processes into a single step, significantly improving inference efficiency.
Weaknesses:
The assumption that the adaptive masking strategy will generalize well to all types of anomalies might be overly optimistic, as some anomalies could still rely on local features.
The computational complexity of training the OmiAD model, especially with the adversarial distillation component, could be a limitation for practical applications with limited resources.

**Questions For Authors:**

1. How would you address scenarios where anomalies are highly dependent on local features, potentially limiting the effectiveness of the adaptive masking strategy?
2. Could you provide more details on the computational overhead of the adversarial distillation process during training, and how it compares to the inference speed improvements?
This is not my area of expertise, so I will be looking closely at other people's comments to adjust the score.

**Relation To Broader Scientific Literature:**

The key contributions of the paper are well-related to the broader scientific literature. The authors discuss how OmiAD advances the field of anomaly detection by addressing the limitations of existing diffusion-based methods. They connect their work to previous research on diffusion models, anomaly detection, and distillation techniques, showing how OmiAD builds upon and improves these approaches. The method's emphasis on reducing shortcut learning and improving inference efficiency aligns with current trends in developing more robust and practical machine learning models.

**Theoretical Claims:**

The theoretical claims regarding the adaptive masking strategy and adversarial score distillation are plausible and well-founded.

---

> ### Author Rebuttal · Authors · 2025-03-31
>
> We truly value the time and effort you invested in carefully reading our paper. Your thoughtful and constructive feedback is highly appreciated.
>
> **Q1:Effectiveness of Adaptive Masking for Localized Anomalies**
>
> A1: Thank you for raising this important point. We agree that anomalies heavily dependent on local features may pose a challenge for global masking strategies. To address this, our adaptive masking mechanism varies the mask ratio with the diffusion step, which allows the model to preserve fine-grained local information at earlier stages and gradually increases difficulty during training. Furthermore, since our model operates on feature-level inputs extracted by EfficientNet, it benefits from strong localized representations.
>
> Additionally, the Real-IAD dataset contains many small-scale anomalies. OmiAD demonstrates excellent performance on this dataset, further validating its effectiveness in handling fine-grained, localized anomalies.
>
> **Q2: Computational overhead of the adversarial distillation**
>
> A2: The computational overhead of adversarial distillation during training is comparable to that of training the teacher model once. This efficiency stems from the fact that the One-step Generator $g_θ$ directly inherits the architecture of the AMDM U-Net (as described in Algorithm 1), and the discriminator shares weights with the one-step generator’s encoder, resulting in minimal additional cost. Moreover, the distillation phase trains the One-step Generator for only 150 epochs—substantially fewer than the 1000 epochs required for AMDM.
>
> At inference time, OmiAD reduces the number of sampling steps from 100 (in AMDM) to just one, leading to a significant speedup.
>
> In summary, adversarial distillation introduces minimal training overhead while enabling substantial inference acceleration through drastic reduction in sampling steps.

---

### Official Review · Reviewer_xWcm · 2025-03-19

**Overall Recommendation:** 4

**Summary:**

The paper proposes a new multi-class anomaly detection method named OmiAD based on diffusion models. First, a diffusion model is trained. Different from standard diffusion models, the images are additionally partially masked to enforce the model to learn the global context. The trained diffusion model is then used by the teacher to distil reconstruction knowledge (using a novel adversarial distillation technique) inside a single-step generation model. It is unclear how the anomaly map is produced, but most likely through a reconstruction difference. OmiAD is then evaluated on four different datasets, achieving SOTA results. The proposed method is also exceptionally fast.

## update after rebuttal

The authors have addressed most of my concerns so I have increased my score to 4.

**Claims And Evidence:**

The paper makes two claims:
- Current diffusion-based anomaly detection models overlook the “identical shortcut” problem of reconstruction-based models, meaning the anomalous regions are not reconstructed to a normal look.
- Current diffusion-based anomaly detection models require multiple steps, which leads to slow inference, which is suboptimal for real-world scenarios.

The paper hypothesizes that a possible solution to the first claim is adaptive masking during the training of the diffusion model. The masking does indeed significantly improve the performance of their model, suggesting that this problem is partially improved. While I know from experience that this is a problem with diffusion-based models, some evidence verifying this would help. For example, by showing the distribution of anomaly scores for previous diffusion-based methods, some anomalous images should have low scores if that is the case. Something similar could be done to showcase that adaptive masking solves this.

The second claim is solved by distilling the diffusion model into a single-step model. The achieved speed is small enough for real-world use and is therefore verified.

**Essential References Not Discussed:**

The paper fails to mention the first approaches to anomaly detection with diffusion models, such as AnoDDPM [5] and DiffAD [6]. While not a big weakness, these methods should at least be mentioned to give a better idea of the development of such methods.

Masking the image has also been done to improve the “identical shortcut” problem. While it has not been as successful, I would at least mention previous methods [7] trying this.

[5] Wyatt, J., Leach, A., Schmon, S. M., & Willcocks, C. G. (2022). Anoddpm: Anomaly detection with denoising diffusion probabilistic models using simplex noise. In Proceedings of the IEEE/CVF conference on computer vision and pattern recognition (pp. 650-656).

[6] Zhang, X., Li, N., Li, J., Dai, T., Jiang, Y., & Xia, S. T. (2023). Unsupervised surface anomaly detection with diffusion probabilistic model. In Proceedings of the IEEE/CVF International Conference on Computer Vision (pp. 6782-6791).

[7] Zavrtanik, V., Kristan, M., & Skočaj, D. (2021). Reconstruction by inpainting for visual anomaly detection. Pattern Recognition, 112, 107706.

**Experimental Designs Or Analyses:**

The experiments verifying the performance in anomaly detection are thorough and adequate. The choice of compared methods is sufficient.

However, I lack a more detailed ablation of some parameters for the model. More specifically, I am interested in how robust the model is to the choice of $p_{min}$, $p_{max}$ and $t_{init}$. I am especially interested in $t_{init}$, as it is set to 960 (at least this value appears in Algorithm 1), which seems incredibly high, meaning the input to the single-step generator is practically noise and very little signal. Additionally, the choice of the feature extractor (EfficientNet) was not ablated.

Other parameters are sufficiently ablated. The inference speed for SimpleNet, however, looks significantly high. From my experience, the model is quite fast and should get a significantly lower inference speed. Additionally, it would help to have the inference speed of the base diffusion model (AMDM) to see the speed improvement brought by distillation.

**Methods And Evaluation Criteria:**

The method is mostly clear, with a few minor details missing: how the anomaly masks are produced (this is the biggest missing detail), what is the architecture of the diffusion model (UNet, DiT?), and what is the input to the one-step generation model - the EfficientNet Features or the input image.

The paper follows the standard evaluation protocol for multi-class anomaly detection methods and uses the standard evaluation metrics. The evaluation protocol, therefore, correctly evaluates anomaly detection performance.

**Other Comments Or Suggestions:**

There are a few typos: Line 358 should probably be Qualitative Results and not Quantitative, Line 257 “distributionclosely” -> “distribution closely”

I would perhaps also add a speed comparison to TransFusion [8] as it requires fewer steps (20) than most diffusion-based models. While I expect the improvement to be 50x or 100x times, it will be a fairer comparison.

Extensive experiment analysis is not a contribution but a scientific standard. I would move it out of the contributions.

[8] Fučka, M., Zavrtanik, V., & Skočaj, D. (2024, September). TransFusion–a transparency-based diffusion model for anomaly detection. In European conference on computer vision (pp. 91-108). Cham: Springer Nature Switzerland.

**Other Strengths And Weaknesses:**

The paper is nicely structured and easy to read. The claims are clearly written and discussed in the paper.

**Questions For Authors:**

I have listed the questions in terms of importance from most important to least important.

- How are the anomaly maps produced?
- How robust is the model to the choice of $t_{init}$? Is there a reason why it is set to 960?
- How is the last step made in Eq. 25 the derivation of the adversarial loss?
- How important is EfficientNet for the model? What happens if you exchange it for some other feature extractor?
- What is the input to the single-step generation model (EfficientNet features or the image)?
- What architecture is used for the diffusion model and the single-step generation model? Is it a UNet, DiT, etc.?
- How is the FPR calculated in the supplementary material?

**Relation To Broader Scientific Literature:**

The paper improves upon previous multi-class diffusion-based models in two aspects: performance and speed. The most important contribution is the speed, making the use of diffusion-based models inside actual industrial scenarios feasible. As shown in Table 2 the model heavily outspeeds current SOTA multiclass diffusion-based methods, DDAD [2] and DiaD [3]. To my knowledge, it is the first method to apply adversarial distillation for diffusion models in the field of anomaly detection. In terms of performance, OmiAD achieves better results than other SOTA methods, such as  HVQ-Trans [4].

[2] Mousakhan, A., Brox, T., & Tayyub, J. (2023). Anomaly detection with conditioned denoising diffusion models. arXiv preprint arXiv:2305.15956.

[3] He, H., Zhang, J., Chen, H., Chen, X., Li, Z., Chen, X., ... & Xie, L. (2024, March). A diffusion-based framework for multi-class anomaly detection. In Proceedings of the AAAI conference on artificial intelligence (Vol. 38, No. 8, pp. 8472-8480).

[4] Lu, R., Wu, Y., Tian, L., Wang, D., Chen, B., Liu, X., & Hu, R. (2023). Hierarchical vector quantized transformer for multi-class unsupervised anomaly detection. Advances in Neural Information Processing Systems, 36, 8487-8500.

**Theoretical Claims:**

The paper has derived the distillation loss. I have checked the derivation, and not all of the steps are entirely clear. For example, in Eq. 25, it is not clear how the final step is achieved and where the minus sign comes from, putting into question whether the loss has been correctly derived.

Additionally, the paper definitions of the $\alpha$ and $\bar{\alpha}$ inside diffusion models are the opposite of the one used in the original paper [1]. I would suggest using the original ones, as this decreases the clarity for readers who are well-versed in diffusion models.

[1] Ho, J., Jain, A., & Abbeel, P. (2020). Denoising diffusion probabilistic models. Advances in neural information processing systems, 33, 6840-6851.

---

> ### Author Rebuttal · Authors · 2025-03-31
>
> We sincerely appreciate the time and effort you invested in carefully reviewing our paper. Your insightful and constructive comments are greatly valued and have helped us improve the clarity and rigor of our work.
>
> **Q1: Inference stage and Anomaly Score Computation**
>
> A1:In the inference stage, we process both normal and anomalous data, following established methods like UniAD and HVQ-Trans for generating anomaly score maps. The pseudocode below outlines this process:
> 1. **Input**: $img$: Input image, $g_θ$: Trained one-step generator, $EfficientNet$: Feature extractor, $t_{init}$: Initial timestep.
> 2. **Output**:  S: Pixel-wise anomaly score map
> 3. **Procedure**:
>    1. **Feature Extraction**: $x_0=EfficientNet(img)$
>    2. **Noising**: $x_t=\sqrt{\bar\alpha_t} \cdot x_0+\sqrt{1 - \bar\alpha_t}\cdot\epsilon,  \quad t=t_{\text{init}},   \epsilon\sim\mathcal{N}(0, I)$
>    3. **One-step Reconstruction**: $\hat{{x}}_0=g_θ(x_t)$
>    4. **Anomaly Score Computation**: $S=\|x_0-\hat{{x}}_0\|_2^2$
>
> **Q2: Ablation study for $t_{init}$ and the rationale for choosing $t_{init}$ = 960**
>
> A2: We conducted ablations with different $t_{init}$ values and found the model remains robust between 800 and 960. We recommend $t_{init} = 960$ for the best trade-off between semantic preservation and anomaly detection.
> |$t_{init}$|600|700|800|940|960|980|1000|
> |-|-|-|-|-|-|-|-|
> |Image AUROC|97.7|98.1|98.3|98.7|98.8|95.9|73.5|
> |Pixel AUROC|96.9|97.1|97.4|97.7|97.7|97.3|80|
>
> We attribute the effective reconstruction at $t_{init} = 960$ to the high similarity among images within the same category. Using a pre-trained EfficientNet for feature extraction improved the model's ability to accurately reconstruct images.
>
> **Q3: Formula 25 derivation**
>
> A3: Thank you very much for your careful derivation. After reviewing it, we found that there was a typographical error. When substituting equation (22) into equation (21), we mistakenly wrote $\bar\beta$ as $\beta$. We have corrected and updated the derivation process, and equation (25) is now as follows:
> $$\mathbb{E}_{q(x\_t\mid x\_g, t)\,p\_\theta(x\_0)}\Big[\Big\langle \epsilon\_\phi(x\_t, t)-\epsilon\_\psi(x\_t, t),\bar{\beta}\_t \nabla\_{x\_t}\log p\_\theta(x\_t)\Big\rangle\Big]$$
>
> $$=-\mathbb{E}_{x\_g \sim p\_\theta(x\_0)x\_t\sim p(x\_t \mid x\_g)} \Big[\Big\langle \epsilon\_\phi(x\_t, t)-\epsilon\_\psi(x\_t, t),\frac{x\_t-\bar{\alpha}\_t x\_g}{\bar{\beta}\_t} \Big\rangle\Big]$$
>
> $$=-\mathbb{E}_{x\_g \sim p\_\theta(x\_0), z,\boldsymbol\epsilon \sim \mathcal{N}(0,I)} \Big[\Big\langle \epsilon\_\phi(x\_t, t)-\epsilon\_\psi(x\_t, t),\epsilon \Big\rangle\Big]\$$
>
> This typographical error does not affect the subsequent loss calculation.
>
> **Q4: Importance of EfficientNet**
>
> A4: ResNet and EfficientNet are commonly used pre-trained feature extractors for anomaly detection. Replacing EfficientNet with ResNet34 on MVTec led to a performance drop (Image/Pixel AUROC: 93.7/96.5). Similar trends were observed in UniAD, where EfficientNet consistently outperformed ResNet. We therefore recommend using EfficientNet for better performance.
>
> **Q5: Input to the One step generator**
>
> A5: The input to the one-step generator is the feature map extracted by EfficientNet.
>
> **Q6: The diffusion model architecture**
>
> A6:  We use a U-Net architecture as the backbone for the diffusion model. To evaluate the generality of our approach, we replaced the U-Net in OmiAD with DiT and U-ViT[1]. The results demonstrate that OmiAD maintains strong performance across different architectures, indicating that our method is architecture-agnostic.
> |Model|DiT|U-ViT|UNet|
> |-|-|-|-|
> |Image AUROC|98.2|98.4|98.8|
> |PIXEL AUROC|97.5|97.4|97.7|
>
> **Q7: Method for Calculating FPR**
>
> A7: In our experiments, the threshold is set based on the number of anomaly samples. We rank the samples by anomaly score and set the threshold to the score ranked at the position matching the number of anomaly samples.
>
> **Q8: More ablation study for $p_{min}$ and $p_{max}$**
>
> A8: We conducted ablation experiments to address this point. For further details, please refer to our response to **reviewer QjD4 Question 4**.
>
> **Q9: Speed of SimpleNet**
>
> A9: We identified that the slow speed was due to the official code transferring data from GPU to CPU and storing it as a numpy array. For further details, please refer to our response to **reviewer 96ub Question 2**.
>
> **Q10: Comparison of speed with Transfusion**
>
> A10: We added a time comparison with TransFusion. Despite fewer steps, TransFusion predicts masks and anomalies at each step and operates at the pixel level rather than in the latent space, both of which limit its inference speed.
> |Dataset|MVTec-ad|VisA|MPDD|Real-IAD|
> |-|-|-|-|-|
> |Inference Time|15.904|15.975|16.002|16.051|
>
> **Due to character limits, we welcome further discussion and are happy to address any remaining questions during the discussion phase.**
>
> [1]Bao F, et al. All are worth words: A vit backbone for diffusion models.

---

> > ### Comment · Reviewer_xWcm · 2025-04-02
> >
> > The authors have satisfactorily answered almost all of my questions, and only one remains:
> >
> > 1. If the one-step generator generates reconstructed EfficientNet features (this assumption is based on the anomaly map generation algorithm), how did you achieve pixel-level reconstructions, as seen in Figure 2?
> >
> > As this is my only concern besides the extreme importance of EfficientNet, I will raise my score.

---

> > > ### Author Response · Authors · 2025-04-02
> > >
> > > We sincerely thank you for the positive feedback and for acknowledging our responses.
> > >
> > > Since EfficientNet and ResNet, which are widely used feature extractors in anomaly detection, are both CNN-based architectures, their convolutional structure enables local feature extraction through kernels while preserving the global spatial relationships of the input. As a result, the relative positions of anomalies in the input image are preserved in the feature space. In other words, the anomaly locations in the feature maps are spatially aligned with their actual positions in the original image. This spatial alignment ensures that anomaly maps generated in the feature space can be reliably used for both anomaly detection and localization. This design choice is consistent with existing works such as UniAD [1] and HVQ-Trans [2].
> > >
> > > Regarding the pixel-level reconstructions shown in Figure 2, we additionally train a decoder to project the EfficientNet features back into the image space. This reconstruction is used solely for visualization purposes and does not participate in the anomaly scoring process. This practice is also consistent with visualization strategies adopted by prior methods such as UniAD[1] and HVQ-Trans[2].
> > >
> > > We are sincerely grateful for your thoughtful review and the time you dedicated to evaluating our work. Your feedback has been invaluable in improving the quality of our paper.
> > >
> > > [1]You, Z, et al. A unified model for multi-class anomaly detection.
> > >
> > > [2]Lu, R, et al. Hierarchical vector quantized transformer for multi-class unsupervised anomaly detection.

---

### Official Review · Reviewer_QjD4 · 2025-03-24

**Overall Recommendation:** 3

**Summary:**

This paper presents OmiAD, a one-step adaptive masked diffusion model designed for multi-class anomaly detection with enhanced inference efficiency.

## Paper contributions:
- The paper introduces an innovative Adaptive Masking Diffusion Model (AMDM) strategy that dynamically adjusts masking patterns based on noise levels. AMDM is proposed to strengthen global context modeling and avoid shortcut reconstruction for anomaly pixels.
- The paper utilizes Adversarial Score Distillation (ASD) to compress multi-step diffusion into single-step inference for test-time efficiency.
- The experimental results show the proposed OmiAD achieves a speed-up over diffusion-based and transformer-based methods on anomaly detection benchmarks (MVTecAD, VisA, MPDD, Real-IAD).

**Claims And Evidence:**

- Details of F-Mask and effectiveness of A-Mask: In Table 3, the authors conduct an ablation study for F-Mask (Fixed Mask) and A-Mask (Adaptive Mask)  to demonstrate the effectiveness of A-Mask. However, there is no introduction to the fixed mask strategy in the methodology section (3.2). How is the mask fixed? Is it fixed over the diffusion timesteps?  What is the choice of probability $p(t)$ for F-Mask?  Besides, the improvement from A-Mask to F-Mask seems incremental.

**Essential References Not Discussed:**

I understand the paper focuses on multi-class AD, but I recommend the authors cite some classical anomaly detection papers: i) training-free methods: SPADE [Niv Cohen, etc], Padim[Thomas Defard, etc], ii) flow-based methods: Cflow-AD[ Denis Gudovskiy], and iii) some diffusion-based AD methods.
Particularly, the training-free method Padim is efficient(>20 fps) and can be directly applied to multi-class cases.

**Experimental Designs Or Analyses:**

The paper lacks a discussion on some important hyper-parameter tunning, such as the choice of $p_{min}, p_{max}$ for Adaptive Mask, the initial timestep $t_{init}$ for one-step generator. Are they sensitive to different types of anomalies?

**Methods And Evaluation Criteria:**

The paper conducts extensive experiments on MVTecAD, VisA, MPDD, Real-IAD. Well-established metrics like Image-level & pixel-level AUROC, F1 max, and AUPRO are used for evaluation.

**Other Comments Or Suggestions:**

Null

**Other Strengths And Weaknesses:**

- The paper does not distinguish between the training stage and the inference stage.
- The writing of the adversarial diffusion distillation part is confusing and not friendly to readers who are not familiar with diffusion distillation.

**Questions For Authors:**

In summary, the most important questions:
1. Computation of the anomaly score
2. Eq.(10), why $x_0$ can be estimated by the masked features $x_m^t$?
3. The paper does not explicitly distinguish between the training stage(only the normal data) and the inference stage(inputs can be normal/abnormal). Most of the method section talks only about how to train the different modules.

**Relation To Broader Scientific Literature:**

It addresses key limitations of prior diffusion-based works, such as multi-class AD, slow inference speed, and shortcut learning.
It lacks comparisons with some classical AD methods, see the "Essential References Not Discussed" section.

**Theoretical Claims:**

I am concerned about the theoretical correctness of the diffusion process since the authors perform two operations i) Gaussian noises and ii) Masking to the features, see eq.(10).  The reconstruction of feature $x_0$ seems okay in the one-step generator, but it is not clear how the diffusion process is affected for the teacher diffusion model(AMDM) training. Especially for eq.(10), why $x_0$ can be estimated by masked features $x_m^t$ with noises? I hope the authors can provide a detailed explanation.

---

> ### Author Rebuttal · Authors · 2025-03-31
>
> We greatly appreciate your thorough review of our paper. Your valuable feedback and constructive suggestions have provided us with a clearer direction for improvement.
>
> **Q1: Inference stage and Anomaly Score Computation**
>
> A1: In the inference stage, we process both normal and anomalous data. The methodology for generating anomaly score maps follows established approaches, such as those outlined in UniAD[1] and HVQ-Trans[2]. The pseudocode below outlines the process for the inference stage, including the generation of anomaly score maps:
> 1. **Input**:  $img$: Original input image, $g_θ$: Trained one-step generator, $EfficientNet$: Feature extractor, $t_{init}$: Initial timestep.
> 2. **Output**: S: Pixel-wise anomaly score map
> 3. **Procedure**:
>    1. **Feature Extraction**: $x_0=EfficientNet(img)$
>    2. **Noising**: $x_t=\sqrt{\bar\alpha_t} \cdot x_0+\sqrt{1 - \bar\alpha_t}\cdot\epsilon,\quad t=t_{\text{init}},  \epsilon\sim\mathcal{N}(0, I)$
>    3. **One step Reconstruction**: $\hat{{x}}_0=g_θ(x_t)$
>    4. **Anomaly Score Computation**: $S=\|x_0-\hat{{x}}_0\|_2^2$
>
> **Q2: Justification for estimating $x_0$ using masked features $x_m^t$**
>
> A2: Anomaly detection requires leveraging global information to reconstruct the anomalous part into the normal modality. We use Equation (10) to predict $x_0$, and apply Equation (11) as a constraint. This formulation allows $\epsilon_\theta$ to compensate for the anomalous part and generate the normal part. We conducted experiments where we replaced $x_m^t$ with $x^t$ in Equation (10). The resulting performance, with Image/Pixel AUROC scores of 90.1/92.9, is inferior to that of AMDM, which achieved 98.4/97.5, thereby further validating our conclusions. In addition, some works, such as DiffMAE[3] and MaskDiT[4], utilize unmasked regions to predict $\hat{{x}}_0$
>
> **Q3: Fixed Mask Strategy**
>
> A3: Unlike A-Mask, where the mask probability varies with each time step, the mask probability in F-Mask remains constant. We set four different mask probabilities [0.1,0.2,0.3,0.4] and conducted experiments, selecting the best-performing combination as the F-Mask strategy for comparison.
>
> **Q4: More ablation study for $p_{min}$, $p_{max}$ and $t_{init}$**
>
> A4: We fixed $p_{min}$=0.1, with varying $p_{max}$, AMDM performance:
> |$p_{max}$|0.3|0.4|0.5|
> |-|-|-|-|
> |Image AUROC|98.1|98.4|98.2|
> |Pixel AUROC|97.4|97.5|97.1|
>
> We fixed $p_{max}$=0.4, with varying $p_{min}$, AMDM performance:
> |$p_{min}$|0|0.1|0.2|
> |-|-|-|-|
> |Image AUROC|98.2|98.4|98.1|
> |Pixel AUROC|97.5|97.5|97.3|
>
> The results show that AMDM maintains stable and strong performance across different $p_{\text{min}}$ and $p_{\text{max}}$ settings. The best performance is observed when $p_{\text{min}}=0.1$ and $p_{\text{max}}=0.4$.
>
> **Ablation study for $t_{init}$**
>
>  We conducted ablation experiments on OmiAD at different $t_{init}$. The experimental results show that the model maintains performance robustness within a wide range of initial time steps (800 to 960). However, when $t_{init}$ = 1000, performance drops significantly due to the noisy input resembling pure noise, which hinders image recovery and reduces anomaly detection ability. We recommend using tinit = 960, as it provides optimal performance at both image and pixel levels.
> | $t_{init}$ |600|700|800|940|960|980|1000|
> |-|-|-|-|-|-|-|-|
> |Image AUROC|97.7|98.1|98.3|98.7|98.8|95.9|73.5|
> |Pixel AUROC|96.9|97.1|97.4|97.7|97.7|97.3|80.0|
>
> **Q5: Handling Large Anomalies**
>
> A5:Thanks to its multi-step generation process and global modeling capability, AMDM is capable of progressively reconstructing missing regions. We visualized AMDM's inference process on samples with missing transistors: before step 860, the model focuses on recovering the missing component, while after step 860, it shifts attention to refining fine-grained image details.
>
> Since OmiAD is distilled from AMDM, it inherits this reconstruction ability and achieves normal restoration from anomalous inputs in a single step.
>
> For pure white or pure black images, where no valid semantic information is available, the model produces meaningless outputs.
>
> **Q6: Essential References Not Discussed**
>
> A6:Thank you for the suggestion. We understand the importance of citing classical anomaly detection methods and will include references to SPADE, Padim, Cflow-AD, and other diffusion-based AD methods in the revised version of the paper.
>
> We also release our single-class anomaly detection results as a public benchmark. For detailed performance, please refer to **Reviewer 96ub, Question 3.**
>
> **Due to character limitations, we would be happy to discuss and address any remaining questions during the discussion phase.**
>
> [1]You, Z, et al. A unified model for multi-class anomaly detection.
>
> [2]Lu, R, et al. Hierarchical vector quantized transformer for multi-class unsupervised anomaly detection.
>
> [3]Wei C, et al. Diffusion models as masked autoencoders.
>
> [4]Zheng H, et al. Fast training of diffusion models with masked transformers.

---

### Decision · Program_Chairs · 2025-05-01

**Decision:**

Accept (poster)

**Comment:**

This paper introduces a one-step adaptive masked diffusion model (AMDM) for multi-class anomaly detection. The AMDM enhances global context modeling and prevents shortcut reconstruction. The proposed method demonstrates a speed-up over existing diffusion-based and transformer-based approaches across multiple anomaly detection benchmarks.

The paper received positive scores from all four reviewers in the initial review phase. The reviewers acknowledged that the authors' rebuttal effectively addressed their concerns, leading them to maintain or even increase their scores, ultimately recommending acceptance. Therefore, the AC recommends accepting this paper.